# M³-Impute: Mask-guided Representation Learning for Missing Value Imputation

## Abstract

Missing values are a common problem that poses significant challenges to data analysis and machine learning. This problem necessitates the development of an effective imputation method to fill in the missing values accurately, thereby enhancing the overall quality and utility of the datasets. Existing imputation methods, however, fall short of considering the 'missingness' information in the data during initialization and modeling the entangled feature and sample correlations explicitly during the learning process, thus leading to inferior performance. We propose M³-Impute, which aims to leverage the missingness information and such correlations with novel masking schemes. M³-Impute first models the data as a bipartite graph and uses an off-the-shelf graph neural network, equipped with a refined initialization process, to learn node embeddings. They are then optimized through M³-Impute's novel feature correlation unit (**FCU**) and sample correlation unit (**SCU**) that enable explicit consideration of feature and sample correlations for imputation. Experiment results on 15 benchmark datasets under three different missing patterns show the effectiveness of M³-Impute by achieving 13 best and 2 second-best MAE scores on average.

## 1 Introduction

Missing values in a dataset are a pervasive issue in real-world data analysis. They arise for various reasons, ranging from the limitations of data collection methods to errors during data transmission and storage. Since many data analysis algorithms cannot directly handle missing values, the most common way to deal with them is to discard the corresponding samples or features with missing values, which would compromise the quality of data analysis. To tackle this problem, missing value imputation algorithms have been proposed to preserve all samples and features by imputing missing values with estimated ones based on the observed values in the dataset, so that the dataset can be analyzed as a complete one without losing any information.

The imputation of missing values usually requires modeling of correlations between different features and samples. Feature-wise correlations help predict missing values from other observed features in the same sample, while sample-wise correlations help predict them in one sample from other similar samples. It is thus important to jointly model the feature-wise and sample-wise correlations in the dataset. In addition, the prediction of missing values also largely depends on the 'missingness' of the data, i.e., whether a certain feature value is observed or not in the dataset. Specifically, the missingness information directly determines which observed feature values can be used for imputation. For example, even if two samples are closely related, it may be less effective to use them for imputation if they have missing values in exactly the same features. It still remains a challenging problem how to jointly model feature-wise and sample-wise correlations with such data missingness.

Among existing methods for missing value imputation, statistical methods [4, 9, 14, 16, 18, 19, 22, 28, 30, 31, 37, 43] extract data correlations with statistical models, which are generally not flexible

in handling mixed data types and struggles to scale up to large datasets. Learning-based imputation methods [10, 24, 27, 29, 33, 42, 50, 51, 53], instead, take advantage of the strong expressiveness and scalability of machine/deep learning algorithms to model data correlations. However, most of them are still built upon the raw tabular data structure as is, which greatly restricts them from jointly modeling the feature-wise and sample-wise correlations. In light of this, graph-based methods [52, 54] have been proposed to model the raw data as a bipartite graph, with samples and features being two different types of nodes. A sample node and a feature node are connected if the feature value is observed in that sample. The missing values are then predicted as the inner product between the embeddings of the corresponding sample and feature nodes. However, this simple prediction does not consider the specific missingness information as mentioned above. For instance, the target feature to impute may have different correlations with features in the samples which have different kinds of missingness; however, the *same* feature-node embedding is still used for their imputation. A similar issue also arises for sample-node embeddings.

In this work, we address these problems by proposing $M^3$-Impute, a mask-guided representation learning method for missing value imputation. The key idea behind $M^3$-Impute is to explicitly utilize the data-missingness information as model input with our proposed novel masking schemes so that it can accurately learn feature-wise and sample-wise correlations in the presence of different kinds of data missingness. $M^3$-Impute first builds a bipartite graph from the data as used in [52]. In the embedding initialization for graph representation learning, however, we not only use the the relationships between samples and their associated features but also the missingness information so as to initialize the embeddings of samples and features jointly and effectively. We then propose novel feature correlation unit (**FCU**) and sample correlation unit (**SCU**) in $M^3$-Impute to explicitly take feature-wise and sample-wise correlations into account for imputation. **FCU** learns the correlations between the target missing feature and observed features within each sample, which are then further updated via a soft mask on the sample missingness information. **SCU** then computes the sample-wise correlations with another soft mask on the missingness information for each pair of samples that have values to impute. We then integrate the output embeddings of **FCU** and **SCU** to estimate the missing values in a dataset. We carry out extensive experiments on 15 open datasets. The results show that $M^3$-Impute outperforms state-of-the-art methods in 13 of the 15 datasets on average under three different settings of missing value patterns, achieving up to $11.47\%$ improvement in MAE compared to the second-best method.

## 2  Related Work

**Statistical methods:** These imputation approaches include joint modeling with expectation-maximization (EM) [9, 16, 22], $k$-nearest neighbors (kNN) [14, 43], and matrix completion [5, 6, 18, 32]. However, joint modeling with EM and matrix completion often lack the flexibility to handle data with mixed modalities, while kNN faces scalability issues due to its high computational complexity. In contrast, $M^3$-Impute is scalable and adaptive to different data distributions.

**Learning-based methods:** Iterative imputation frameworks [1, 2, 15, 20, 23, 24, 35, 41, 44, 45], such as MICE [45] and HyperImpute [23], have been extensively studied. These iterative frameworks apply different imputation methods for each feature and iteratively estimate missing values until convergence. In addition, for deep neural network learners, both generative models [27, 29, 36, 50, 51, 53], such as GAIN [50] and MIWAE [29], and discriminative models [10, 24, 48], such AimNet [48], have also been proposed. However, these methods are built upon raw tabular data structures, which fall short of capturing the complex correlations in features, samples, and their combination [54]. In contrast, $M^3$-Impute is based on the bipartite graph modeling of the data, which is more suitable for learning the data correlations for imputation.

**Graph neural network-based methods:** GNN-based methods [40, 52, 54] are proposed to address the drawbacks mentioned above due to their effectiveness in modeling complex relations between entities. Among them, GRAPE [52] transforms tabular data into a bipartite graph where features are one type of node and samples are the other. A sample node is connected to a feature node only if the corresponding feature value is present. This transformation allows the imputation task to be framed as a link prediction problem, where the inner product of the learned node embeddings is computed as the predicted values. IGRM [54] further enhances the bipartite graph by explicitly introducing linkages between sample nodes to facilitate message propagation between samples. However, these methods do not effectively encode the missingness information of different samples and features into

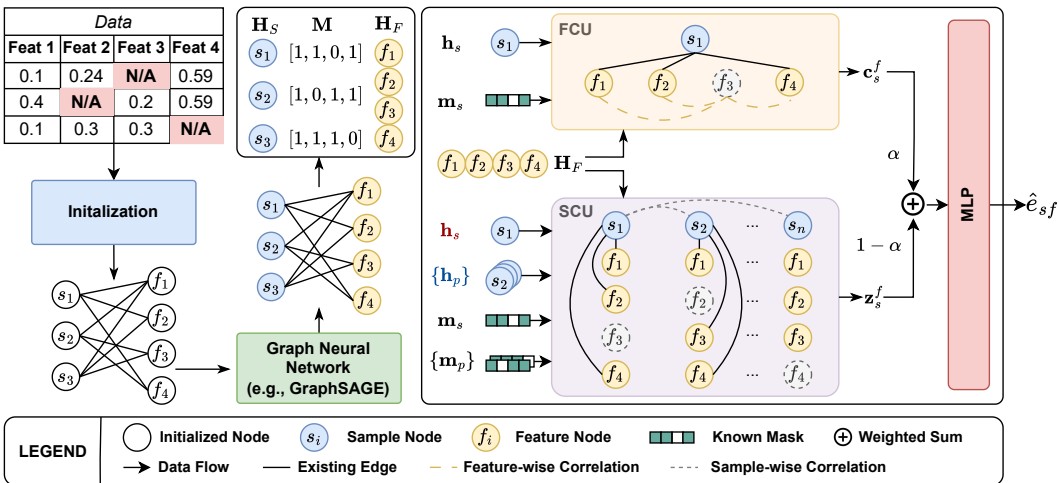

Figure 1: Overview of the M³-Impute model.

the imputation process, which can impair their imputation accuracy. In contrast, M³-Impute enables explicit modeling of missingness information through novel masking schemes so that feature-wise and sample-wise correlations can be accurately captured in the imputation process.

# 3  M³-Impute

## 3.1  Overview

We here provide an overview of M³-Impute to impute the missing value of feature $f$ for a given sample $s$, as depicted in Figure 1. Initially, the data matrix with missing values is modeled as an undirected bipartite graph, and the missing value is imputed by predicting the edge weight $\hat{e}_{sf}$ of its corresponding missing edge (Section 3.2). M³-Impute next employs a GNN model, such as GraphSAGE [17], on the bipartite graph to learn the embeddings of samples and features. These embeddings, along with the known masks of the data matrix (used to indicate which feature values are available in each sample), are then input into our novel feature correlation unit (**FCU**) and sample correlation unit (**SCU**), which shall be explained in Section 3.3 and Section 3.4, to obtain feature-wise and sample-wise correlations, respectively. Finally, M³-Impute takes the feature-wise and sample-wise correlations into a multi-layer perceptron (MLP) to predict the missing feature value $\hat{e}_{sf}$ (Section 3.5). The whole process, including the embedding generation, is trained in an end-to-end manner.

## 3.2  Initialization Unit

Let $\mathbf{A} \in \mathbb{R}^{n \times m}$ be an $n \times m$ matrix that consists of $n$ data samples and $m$ features, where $\mathbf{A}_{ij}$ denotes the $j$-th feature value of the $i$-th data sample. We introduce an $n \times m$ mask matrix $\mathbf{M} \in \{0, 1\}^{n \times m}$ for $\mathbf{A}$ to indicate that the value of $\mathbf{A}_{ij}$ is *observed* when $\mathbf{M}_{ij} = 1$. In other words, the goal of imputation here is to predict the missing feature values $\mathbf{A}_{ij}$ for $i$ and $j$ such that $\mathbf{M}_{ij} = 0$. We define the *masked* data matrix $\mathbf{D}$ to be $\mathbf{D} = \mathbf{A} \odot \mathbf{M}$, where $\odot$ is the Hadamard product, i.e., the element-wise multiplication of two matrices.

As used in recent studies [52, 54], we model the masked data matrix $\mathbf{D}$ as a bipartite graph and tackle the missing value imputation problem as a link prediction task on the bipartite graph. Specifically, $\mathbf{D}$ is modeled as an undirected bipartite graph $\mathcal{G} = (\mathcal{S} \cup \mathcal{F}, \mathcal{E})$, where $\mathcal{S} = \{s_1, s_2, \ldots, s_n\}$ is the set of 'sample' nodes and $\mathcal{F} = \{f_1, f_2, \ldots, f_m\}$ is the set of 'feature' nodes. Also, $\mathcal{E}$ is the set of edges that only exist between sample node $s$ and feature node $f$ when $\mathbf{D}_{sf} \neq 0$, and each edge $(s, f) \in \mathcal{E}$ is associated with edge weight $e_{sf}$, which is given by $e_{sf} = \mathbf{D}_{sf}$. Then, the missing value imputation problem becomes, for any missing entries in $\mathbf{D}$ (where $\mathbf{D}_{sf} = 0$), to predict their corresponding edge weights by developing a learnable mapping $F(\cdot)$, i.e.,

$$\hat{e}_{sf} = F(\mathcal{G}, (s, f) \notin \mathcal{E}). \tag{1}$$

The recent studies that use the bipartite graph modeling [52, 54] initialize all sample node embeddings as all-one vectors and feature node embeddings as one-hot vectors, which have a value 1 in the positions representing their respective features and 0's elsewhere. We observe, however, that such an initialization does not effectively utilize the information from the masked data matrix, which leads to inferior imputation accuracy, as shall be demonstrated in Section 4.3. Thus, in $M^3$-Impute, we propose to initialize each sample node embedding based on its associated (initial) feature embeddings instead of initializing them separately. While the feature embeddings are randomly initialized, the sample node embeddings are initialized in a way that reflects the embeddings of the features whose values are available in their corresponding samples.

Let $\mathbf{h}_f^0$ be the initial embedding of feature $f$, which is a randomly initialized $d$-dimensional vector, and define $\mathbf{H}_F^0 = [\mathbf{h}_{f_1}^0 \mathbf{h}_{f_2}^0 \dots \mathbf{h}_{f_m}^0] \in \mathbb{R}^{d \times m}$. Also, let $\mathbf{d}_s \in \mathbb{R}^m$ be the $s$-th column vector of $\mathbf{D}^\top$, which is a vector of the feature values of sample $s$, and let $\mathbf{m}_s \in \mathbb{R}^m$ be its corresponding mask vector, i.e., $\mathbf{m}_s = \mathrm{col}_s(\mathbf{M}^\top)$, where $\mathrm{col}_s(\cdot)$ denotes the $s$-th column vector of the matrix. We then initialize the embedding $\mathbf{h}_s^0$ of each sample node $s$ as follows:

$$\mathbf{h}_s^0 = \phi\Big(\mathbf{H}_F^0 \big[\mathbf{d}_s + \epsilon(\mathbb{1} - \mathbf{m}_s)\big]\Big), \tag{2}$$

where $\mathbb{1} \in \mathbb{R}^m$ is an all-one vector, and $\phi(\cdot)$ is an MLP. Note that the term $\mathbf{d}_s + \epsilon(\mathbb{1} - \mathbf{m}_s)$ indicates a vector that consists of observable feature values of $s$ and some small positive values $\epsilon$ in the places where the feature values are unavailable (masked out).

## 3.3 Feature Correlation Unit

To improve the accuracy of missing value imputation, we aim to fully exploit feature correlations which often appear in the datasets. While the feature correlations are naturally captured by GNNs, we observe that there is still room for improvement. We propose **FCU** as an integral component of $M^3$-Impute to fully exploit the feature correlations.

To impute the missing value of feature $f$ for a given sample $s$, **FCU** begins by computing the feature 'context' vector of sample $s$ in the embedding space that reflects the correlations between the target missing feature $f$ and observed features. Let $\mathbf{h}_f \in \mathbb{R}^d$ be the learned embedding vector of feature $f$ from the GNN, and let $\mathbf{H}_F$ be the $d \times m$ matrix that consists of all the learned feature embedding vectors. We first obtain dot-product similarities between feature $f$ and all the features in the embedding space, i.e., $\mathbf{H}_F^\top \mathbf{h}_f$. We then mask out the similarity values with respect to *non-observed* features in sample $s$. Here, instead of applying the mask vector $\mathbf{m}_s$ of sample $s$ directly, we use a learnable 'soft' mask vector, denoted by $\mathbf{m}_s'$, which is defined to be $\mathbf{m}_s' = \sigma_1(\mathbf{m}_s) \in \mathbb{R}^m$, where $\sigma_1(\cdot)$ is an MLP with the GELU activation function [21]. In other words, we obtain feature-wise similarities with respect to sample $s$, denoted by $\mathbf{r}_s^f$, as follows:

$$\mathbf{r}_s^f = \sigma_2\big((\mathbf{H}_F^\top \mathbf{h}_f) \odot \mathbf{m}_s'\big) \in \mathbb{R}^d, \tag{3}$$

where $\sigma_2(\cdot)$ denotes another MLP with the GELU activation function. **FCU** next obtains the Hadamard product between the learned embedding vector of sample $s$, $\mathbf{h}_s$, and the feature-wise similarities with respect to sample $s$, $\mathbf{r}_s^f$, to learn their joint representations in a multiplicative manner. Specifically, **FCU** obtains the feature context vector of sample $s$, denoted by $\mathbf{c}_s^f$, as follows:

$$\mathbf{c}_s^f = \sigma_3\big(\mathbf{h}_s \odot \mathbf{r}_s^f\big) \in \mathbb{R}^d, \tag{4}$$

where $\sigma_3(\cdot)$ is also an MLP with the GELU activation function. That is, **FCU** fuses the representation vector of $s$ and the vector that has embedding similarity values between the target feature $f$ and the available features in $s$ through the effective use of the soft mask $\mathbf{m}_s'$. From (3) and (4), the operations of **FCU** can be written as

$$\mathbf{c}_s^f = \mathbf{FCU}(\mathbf{h}_s, \mathbf{m}_s, \mathbf{H}_F) = \sigma_3\big(\mathbf{h}_s \odot \sigma_2\big((\mathbf{H}_F^\top \mathbf{h}_f) \odot \sigma_1(\mathbf{m}_s)\big)\big). \tag{5}$$

## 3.4 Sample Correlation Unit

To measure similarities between $s$ and other samples, a common approach would be to use the dot product or cosine similarity between their embedding vectors. This approach, however, fails to take into account the observability or availability of each feature in a sample. It also does

169 not capture the fact that different observed features are of different importance to the target fea-
170 ture to impute when it comes to measuring the similarities. We introduce **SCU** as another inte-
171 gral component of M³-Impute to compute the sample 'context' vector of sample $s$ by incorpo-
172 rating the embedding vectors of its similar samples as well as different weights of observed fea-
173 tures. **SCU** works based on the two novel masking schemes, which shall be explained shortly.
174

175 Suppose we are to impute the missing value of feature $f$ for a given sample
176 $s$. **SCU** aims to leverage the information from the samples that are similar
177 to $s$. As a first step to this end, we create a subset of samples $\mathcal{P} \subset \mathcal{S}$ that
178 are similar to $s$. Specifically, we randomly choose and put a sample into
179 $\mathcal{P}$ with probability that is proportional to the cosine similarity between $s$
180 and the sample. This operation is repeated without replacement until $\mathcal{P}$
181 reaches a given size.

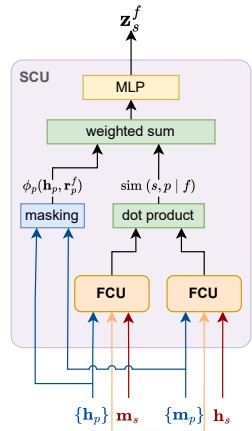

Figure 2: **SCU**.

182 **Mutual Sample Masking:** Given a subset of samples $\mathcal{P}$ that include $s$,
183 we first compute the pairwise similarities between $s$ and other samples in
184 the subset $\mathcal{P}$. While they are computed in a similar way to **FCU**, we only
185 consider the commonly observed features (or the common ones that have
186 feature values) in both $s$ and its peer $p \in \mathcal{P} \setminus \{s\}$, to calculate their pair-
187 wise similarity in the sense that the missing value of feature $f$ is inferred.
188 Specifically, we compute the pairwise similarity between $s$ and $p \in \mathcal{P} \setminus \{s\}$,
189 which is denoted by $\text{sim}(s, p \mid f)$, as follows:

$$\text{sim}(s, p \mid f) = \mathbf{FCU}(\mathbf{h}_s, \mathbf{m}_p, \mathbf{H}_f) \cdot \mathbf{FCU}(\mathbf{h}_p, \mathbf{m}_s, \mathbf{H}_f) \in \mathbb{R}, \quad (6)$$

190 where $\mathbf{h}_s$ and $\mathbf{h}_p$ are the learned embedding vectors of samples $s$ and $p$ from the GNN, respectively,
191 and $\mathbf{m}_s$ and $\mathbf{m}_p$ are their respective mask vectors. Note that the multiplication in the RHS of (6) is
192 the dot product.

193 **Irrelevant Feature Masking:** After we obtain the pairwise similarities between $s$ and other samples
194 in $\mathcal{P}$, it would be natural to consider a weighted sum of their corresponding embedding vectors, i.e.,
195 $\sum_{p \in \mathcal{P} \setminus \{s\}} \text{sim}(s, p \mid f) \, \mathbf{h}_p$, in imputing the value of the target feature $f$. However, we observe that
196 $\mathbf{h}_p$ contains the information from the features whose values are available in $p$ as well as possibly
197 other features as it is learned via the so-called neighborhood aggregation mechanism that is central
198 to GNNs, but some of the features may be irrelevant in inferring the value of feature $f$. Thus, instead
199 of using $\{\mathbf{h}_p\}$ directly, we introduce a $d$-dimensional mask vector $\mathbf{r}_p^f$ for $\mathbf{h}_p$, which is to mask out
200 potentially irrelevant feature information in $\mathbf{h}_p$, when it comes to imputing the value of feature $f$.
201 Specifically, it is defined by

$$\mathbf{r}_p^f = \sigma_4 \left([\mathbf{m}_p; \overline{\mathbf{m}}_f]\right) \in \mathbb{R}^d, \quad (7)$$

202 where $\overline{\mathbf{m}}_f$ is an $m$-dimensional one-hot vector that has a value 1 in the place of feature $f$ and 0's
203 elsewhere, $[\cdot\,;\cdot]$ denotes the vector concatenation operation, and $\sigma_4(\cdot)$ is an MLP with the GELU
204 activation function. Note that the rationale behind the design of $\mathbf{r}_p^f$ is to embed the information on
205 the features whose values are present in $p$ as well as the information on the target feature $f$ to impute.
206 The mask $\mathbf{r}_p^f$ is then applied to $\mathbf{h}_p$ to obtain the masked embedding vector of $p$ as follows:

$$\phi_p(\mathbf{h}_p, \mathbf{r}_p^f) = \sigma_5 \left(\mathbf{h}_p \odot \mathbf{r}_p^f\right) \in \mathbb{R}^d, \quad (8)$$

207 where $\sigma_5(\cdot)$ is also an MLP with the GELU activation function. Once we have the masked embed-
208 ding vectors of samples (excluding $s$) in $\mathcal{P}$, we finally compute the sample context vector of sample
209 $s$, denoted by $\mathbf{z}_s^f$, which is a weighted sum of the masked embedding vectors with weights being the
210 pairwise similarity values, i.e.,

$$\mathbf{z}_s^f = \sigma_6 \left( \sum_{p \in \mathcal{P} \setminus \{s\}} \text{sim}(s, p \mid f) \, \phi_p(\mathbf{h}_p, \mathbf{r}_p^f) \right) \in \mathbb{R}^d, \quad (9)$$

211 where $\sigma_6(\cdot)$ is again an MLP with the GELU activation function. From (6)–(9), the operations of
212 **SCU** can be written as

$$\mathbf{z}_s^f = \mathbf{SCU}(\mathbf{H}_{\mathcal{P}}, \mathbf{M}_{\mathcal{P}}, \mathbf{H}_F) = \sigma_6 \left( \sum_{p \in \mathcal{P} \setminus \{s\}} \text{sim}(s, p \mid f) \, \sigma_5 \left(\mathbf{h}_p \odot \sigma_4 \left([\mathbf{m}_p; \overline{\mathbf{m}}_f]\right)\right) \right), \quad (10)$$

213 where $\mathbf{H}_{\mathcal{P}} = \{\mathbf{h}_p, p \in \mathcal{P}\}$ and $\mathbf{M}_{\mathcal{P}} = \{\mathbf{m}_p, p \in \mathcal{P}\}$.

---
**Algorithm 1** Forward computation of $M^3$-Impute to impute the value of feature $f$ for sample $s$.
---
1: **Input:** Bipartite graph $\mathcal{G}$, initial feature node embeddings $\mathbf{H}_F^0$, GNN model (e.g., GraphSAGE) $\mathbf{GNN}(\cdot)$, known mask matrix $\mathbf{M}$, and a subset of samples $\mathcal{P} \subset \mathcal{S}$.
2: **Output:** Predicted missing feature value $\hat{e}_{sf}$.
3: Obtain initial sample node embeddings $\mathbf{H}_S^0$ according to Equation (2).
4: $\mathbf{H}_S, \mathbf{H}_F = \mathbf{GNN}(\mathbf{H}_S^0, \mathbf{H}_F^0, \mathcal{G})$.              ▷ Perform graph representation learning
5: $\mathbf{c}_s^f = \mathbf{FCU}(\mathbf{h}_s, \mathbf{m}_s, \mathbf{H}_F)$.
6: $\mathbf{z}_s^f = \mathbf{SCU}(\mathbf{H}_{\mathcal{P}}, \mathbf{M}_{\mathcal{P}}, \mathbf{H}_F)$.
7: Predict the missing feature value $\hat{e}_{sf}$ using Equation (11).
---

## 3.5 Imputation

For a given sample $s$, to impute the missing value of feature $f$, $M^3$-Impute obtains its feature context vector $\mathbf{c}_s^f$ and sample context vector $\mathbf{z}_s^f$ through **FCU** and **SCU**, respectively, which are then used for imputation. Specifically, it is done by predicting the corresponding edge weight $\hat{e}_{sf}$ as follows:

$$\hat{e}_{sf} = \phi_\alpha \left( (1-\alpha)\mathbf{c}_s^f + \alpha \mathbf{z}_s^f \right), \tag{11}$$

where $\phi_\alpha(\cdot)$ denotes an MLP with a non-linear activation function (i.e., ReLU for continuous values and softmax for discrete ones), and $\alpha$ is a learnable scalar parameter. This scalar parameter $\alpha$ is introduced to strike a balance between leveraging feature-wise correlation and sample-wise correlation. It is necessary because the quality of $\mathbf{z}_s^f$ relies on the quality of the samples chosen in $\mathcal{P}$, so overly relying on $\mathbf{z}_s^f$ would backfire if their quality is not as desired. To address this problem, instead of employing a fixed weight $\alpha$, we make $\alpha$ learnable and adaptive in determining the weights for $\mathbf{c}_s^f$ and $\mathbf{z}_s^f$. Note that this kind of learnable parameter approach has been widely adopted in natural language processing [26, 34, 38, 46] and computer vision [8, 55, 56], showing superior performance to its fixed counterpart. In $M^3$-Impute, the scalar parameter $\alpha$ is learned based on the similarity values between $s$ and its peer samples $p \in \mathcal{P} \setminus \{s\}$ as follows:

$$\alpha = \phi_\gamma \Big( \mathop{\Big\|}_{p \in \mathcal{P} \setminus \{s\}} \text{sim}\,(s, p \mid f) \Big), \tag{12}$$

where $\|$ represents the concatenation operation, and $\phi_\gamma(\cdot)$ is an MLP with the activation function $\gamma(x) = 1 - 1/e^{|x|}$. The overall operation of $M^3$-Impute is summarized in Algorithm 1. To learn network parameters, we use cross-entropy loss and mean square error loss for imputing discrete and continuous feature values, respectively.

# 4 Experiments

## 4.1 Experiment Setup

**Datasets:** We conduct experiments on 15 open datasets. These real-world datasets consist of mixed data types with both continuous and discrete values and cover different domains including civil engineering (CONCRETE, ENERGY), physics and chemistry (YACHT), thermal dynamics (NAVAL), etc. Since the datasets are fully observed, we introduce missing values by applying a randomly generated mask to the data matrix. Specifically, as used in prior studies [23, 24], we apply three masking generation schemes, namely missing completely at random (MCAR), missing at random (MAR), and missing not at random (MNAR).[1] We use MCAR with a missing ratio of 30%, unless otherwise specified. We follow the preprocessing steps adopted in [52, 54] to scale feature values to [0, 1] with a MinMax scaler [25]. Due to the space limit, we below present the results of eight datasets that are used in Grape [52] and report the other results in Appendix.

**Baseline models:** $M^3$-Impute is compared against popular and state-of-the-art imputation methods, including statistical methods, deep generative methods, and graph-based methods listed as follows: **MEAN**: It imputes the missing value $\hat{e}_{sf}$ as the mean of observed values in feature $f$ from all the samples. K-nearest neighbors (**kNN**) [43]: It imputes the missing value $\hat{e}_{sf}$ using the kNNs that have observed values in feature $f$ with weights that are based on the Euclidean distance to sample $s$. Multivariate imputation by chained equations (**Mice**) [45]: This method runs multiple regressions where each missing value is modeled upon the observed non-missing values. Iterative

---
[1]More details about the datasets and mask generation for missing values can be found in Appendix.

Table 1: Imputation accuracy in MAE. MAE scores are enlarged by 10 times.

| | Yacht | Wine | Concrete | Housing | Energy | Naval | Kin8nm | Power |
|---|---|---|---|---|---|---|---|---|
| Mean | 2.09 | 0.98 | 1.79 | 1.85 | 3.10 | 2.31 | 2.50 | 1.68 |
| Svd [18] | 2.46 | 0.92 | 1.94 | 1.53 | 2.24 | 0.50 | 3.67 | 2.33 |
| Spectral [30] | 2.64 | 0.91 | 1.98 | 1.46 | 2.26 | 0.41 | 2.80 | 2.13 |
| Mice [45] | 1.68 | 0.77 | 1.34 | 1.16 | 1.53 | 0.20 | 2.50 | 1.16 |
| kNN [43] | 1.67 | 0.72 | 1.16 | 0.95 | 1.81 | 0.10 | 2.77 | 1.38 |
| Gain [50] | 2.26 | 0.86 | 1.67 | 1.23 | 1.99 | 0.46 | 2.70 | 1.31 |
| Miwae [29] | 4.68 | 1.00 | 1.81 | 3.81 | 2.79 | 2.37 | 2.57 | 1.74 |
| Grape [52] | 1.46 | **0.60** | 0.75 | 0.64 | 1.36 | 0.07 | 2.50 | 1.00 |
| Miracle [24] | 42.97 | 1.13 | 1.71 | 42.23 | 41.43 | 0.17 | **2.49** | 1.15 |
| HyperImpute [23] | 1.76 | 0.67 | 0.84 | 0.82 | **1.32** | **0.04** | 2.58 | 1.06 |
| M$^3$-Impute | **1.33** | **0.60** | **0.71** | **0.60** | **1.32** | 0.06 | 2.50 | **0.99** |

SVD (**Svd**) [18]: It imputes missing values by solving a matrix completion problem with iterative low-rank singular value decomposition. Spectral regularization algorithm (**Spectral**) [30]: This matrix completion algorithm uses the nuclear norm as a regularizer and imputes missing values with iterative soft-thresholded SVD. **Miwae** [29]: It works based on an autoencoder generative model trained to maximize a potentially tight lower bound of the log-likelihood of the observed data and Monte Carlo techniques for imputation. **Miracle** [24]: It uses the imputation results from naive methods such as MEAN and refines them iteratively by learning a missingness graph (m-graph) and regularizing an imputation function. **Gain** [50]: This method trains a data imputation generator with a generalized generative adversarial network in which the discriminator aims to distinguish between real and imputed values. **Grape** [52]: It models the data as a bipartite graph and imputes missing values by predicting the weights of the missing edges, each of which is done based on the inner product between the embeddings of its corresponding sample and feature nodes. **HyperImpute** [23]: HyperImpute is a framework that conducts an extensive search among a set of imputation methods, selecting the optimal imputation method with fine-tuned parameters for each feature in the dataset.

**Model configurations:** Parameters of M$^3$-Impute are updated by the Adam optimizer with a learning rate of 0.001 for 40,000 epochs. For graph representation learning, we use a variant of Graph-SAGE [17], which not only learns node embeddings but also edge embeddings via the neighborhood aggregation mechanism, as similarly used in [52]. We consider its three-layer GNN model. We employ mean-pooling as the aggregation function and use ReLU as the activation function for the GNN layers. We set the embedding dimension $d$ to 128. It is known that randomly dropping out a subset of observable edges during training improves the model's generalization ability. We also leverage the observation and randomly drop $50\%$ of observable edges during training. For each experiment, we conduct five runs with different random seeds and report the average results.

### 4.2 Overall Performance

We first compare the feature imputation performance of M$^3$-Impute with popular and state-of-the-art imputation methods. As shown in Table 1, M$^3$-Impute achieves the lowest imputation MAE for six out of the eight examined datasets and the second-best MAE scores in the other two, which validates the effectiveness of M$^3$-Impute. For KIN8NM dataset, M$^3$-Impute underperforms Miracle. It is mainly because each feature in KIN8NM is independent of the others, so none of the observed features can help impute missing feature values. For NAVAL dataset, the only model that outperforms M$^3$-Impute is HyperImpute [23]. In the NAVAL dataset, nearly every feature exhibits a strong linear correlation with the other features, i.e., every pair of features has correlation coefficient close to one. This allows HyperImpute to readily select a linear model from its model pool for each feature to impute. Nonetheless, M$^3$-Impute exhibits overall superior performance to the baselines as it can be well adapted to each dataset that possesses different amounts of correlations over features and samples. In other words, M$^3$-Impute benefits from explicitly incorporating feature-wise and sample-wise correlations together with our carefully designed mask schemes. Furthermore, we evaluate the performance of M$^3$-Impute under MAR and MNAR settings. We observe that M$^3$-Impute consistently outperforms all the baselines under all datasets and achieves a larger margin in the improvement compared to the case with MCAR setting. This implies that M$^3$-Impute is also effective in handling different patterns of missing values in the input data. Comprehensive results are provided in Appendix.

Table 2: Ablation study. $M^3$-Uniform stands for $M^3$-Impute with the uniform sampling strategy.

| | Yacht | Wine | Concrete | Housing | Energy | Naval | Kin8nm | Power |
|---|---|---|---|---|---|---|---|---|
| HyperImpute | $1.76 \pm .03$ | $0.67 \pm .01$ | $0.84 \pm .02$ | $0.82 \pm .01$ | $1.32 \pm .02$ | $\mathbf{0.04} \pm .00$ | $2.58 \pm .05$ | $1.06 \pm .01$ |
| Grape | $1.46 \pm .01$ | $\mathbf{0.60} \pm .00$ | $0.75 \pm .01$ | $0.64 \pm .01$ | $1.36 \pm .01$ | $0.07 \pm .00$ | $\mathbf{2.50} \pm .00$ | $1.00 \pm .00$ |
| Architecture | | | | | | | | |
| Init Only | $1.43 \pm .01$ | $\mathbf{0.60} \pm .00$ | $0.74 \pm .00$ | $0.63 \pm .01$ | $1.35 \pm .01$ | $0.06 \pm .00$ | $\mathbf{2.50} \pm .00$ | $\mathbf{0.99} \pm .00$ |
| Init+**FCU** | $1.35 \pm .01$ | $0.61 \pm .00$ | $0.72 \pm .03$ | $0.61 \pm .02$ | $1.32 \pm .00$ | $0.07 \pm .01$ | $\mathbf{2.50} \pm .00$ | $\mathbf{0.99} \pm .00$ |
| Init+**SCU** | $1.37 \pm .01$ | $\mathbf{0.60} \pm .00$ | $0.73 \pm .00$ | $0.63 \pm .01$ | $\mathbf{1.30} \pm .00$ | $0.09 \pm .01$ | $\mathbf{2.50} \pm .00$ | $1.00 \pm .00$ |
| $M^3$-Impute | $\mathbf{1.33} \pm .04$ | $\mathbf{0.60} \pm .00$ | $\mathbf{0.71} \pm .01$ | $\mathbf{0.60} \pm .00$ | $1.32 \pm .01$ | $0.06 \pm .00$ | $\mathbf{2.50} \pm .00$ | $\mathbf{0.99} \pm .00$ |
| Sampling Strategy | | | | | | | | |
| $M^3$-Uniform | $1.34 \pm .01$ | $\mathbf{0.60} \pm .00$ | $0.73 \pm .01$ | $0.61 \pm .00$ | $1.31 \pm .00$ | $0.06 \pm .00$ | $\mathbf{2.50} \pm .00$ | $\mathbf{0.99} \pm .00$ |

## 4.3 Ablation Study

To study the effectiveness of three integral components of $M^3$-Impute, we consider three variants of $M^3$-Impute, each with a subset of the components, namely initialization only (Init Only), initialization + **FCU** (Init + **FCU**), and initialization + **SCU** (Init + **SCU**). The performance of these variants are evaluated against the top-performing imputation baselines such as Grape and HyperImpute. As shown in Table 2, the three variants derived from $M^3$-Impute achieve lower MAE values than both baselines in most datasets, demonstrating the effectiveness of our novel components in $M^3$-Impute.

Specifically, for initialization only, the key difference between $M^3$-Impute and Grape lies in our refined initialization process of feature-node and sample-node embeddings. The reduced MAE values observed by the Init Only variant demonstrate that our proposed initialization process is more effective in utilizing information between samples and their associated features, including missing ones, as compared to the basic initialization used in [52]. In addition, we observe that when **FCU** or **SCU** is incorporated, MAE values are further reduced for most datasets. This validates that explicitly modeling feature-wise or sample-wise correlations through our novel masking schemes can improve imputation accuracy. When all the three components are combined together as in $M^3$-Impute, they work synergistically to lower MAE values, validating the efficacy of explicit consideration of both sample-wise and feature-wise correlations (in addition to the refined initialization process) for missing data imputation.

## 4.4 Robustness

**Missing ratio:** In practice, datasets may possess different missing ratios. To validate the model's robustness under such circumstances, we evaluate the performance of $M^3$-Impute and other baseline models with varying missing ratios, i.e., 0.1, 0.3, 0.5, and 0.7. Figure 3 shows their performance. We use the MAE of HyperImpute ($HI$) as the reference performance and offset the performance of each model by $\mathrm{MAE}_x - \mathrm{MAE}_{HI}$, where $x$ represents the considered model. For clarity, we here only report the results of four top-performing models. As shown in Figure 3, $M^3$-Impute outperforms other baseline models for almost all the cases, especially under YACHT, CONCRETE, ENERGY, and HOUSING datasets. It is worth noting that modeling feature correlations in these datasets is particularly challenging due to the presence of considerable amounts of weakly correlated features, along with a few strongly correlated ones. Nonetheless, **FCU** and **SCU** in $M^3$-Impute were able to better capture such correlations with our efficient masking schemes, thereby resulting in a large improvement in imputation accuracy. In addition, for KIN8NM dataset, $M^3$-Impute ties with the second-best model, Grape. As mentioned in Section 4.2, each feature in KIN8NM is independent of the others, so none of the observed features can help impute missing feature values. For NAVAL dataset, where each feature strongly correlates with the others, $M^3$-Impute surpasses Grape but falls short of HyperImpute, due to the same reason as discussed above. Overall, $M^3$-Impute is robust to various missing ratios. Comprehensive results for all the baseline models can be found in Appendix.

**Sampling strategy in SCU:** While **SCU** uses a sampling strategy based on pairwise cosine similarities to construct a subset of samples $\mathcal{P}$, the simplest sampling strategy to build $\mathcal{P}$ would be to choose samples uniformly at random without replacement ($M^3$-Uniform). Intuitively, this approach cannot identify similar peer samples accurately and thus would lead to inferior performance. Nonetheless, as shown in Table 2, even with this naive uniform sampling strategy, $M^3$-Uniform still outperforms the two leading imputation baselines.

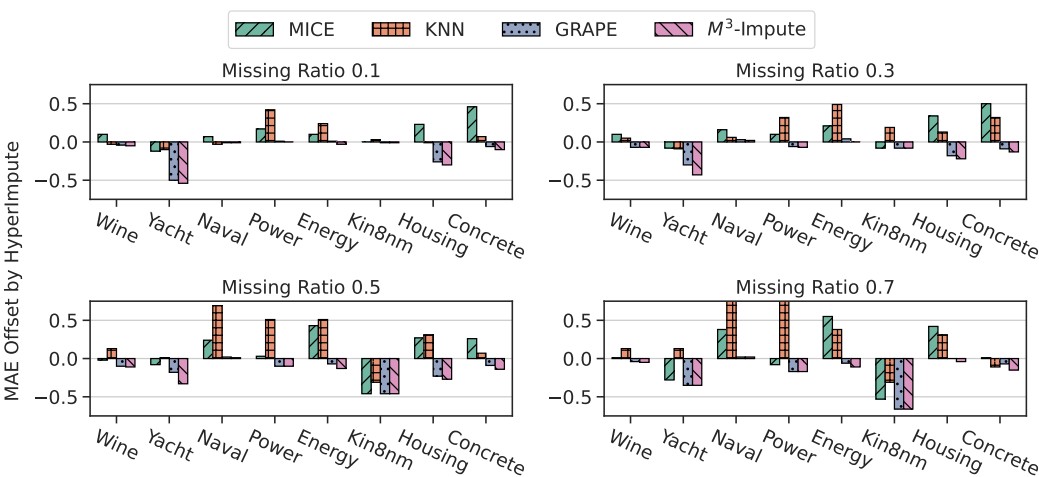

Figure 3: Model performance vs. missing ratios. MAE scores are offset by HyperImpute [23].

**Size of $\mathcal{P}$ in SCU:** Intuitively, neither an excessively small nor overly large size of the sample subset $\mathcal{P}$ is optimal. Too few peer samples leave **SCU** with insufficient information to learn sample-wise correlations, while too many peer samples may include quite a few dissimilar ones, which may introduce significant noise to the computation of **SCU** and thus degrade the performance. Table 3 shows the performance of M³-Impute with varying numbers of peer samples. In general, the trends agree with our intuition. Although the optimal size varies across different datasets, we observe that having the number of peer samples to be 5 to 10 achieves the overall best imputation accuracy.

Table 3: MAE scores for varying peer-sample size ($|\mathcal{P}|-1$) and different values of $\epsilon$.

|  | Yacht | Wine | Concrete | Housing | Energy | Naval | Kin8nm | Power |
|---|---|---|---|---|---|---|---|---|
| Peer = 1 | 1.34 ± .00 | **0.60** ± .00 | 0.73 ± .00 | 0.61 ± .01 | 1.32 ± .00 | **0.06** ± .00 | **2.5** ± .00 | **0.99**± .00 |
| Peer = 2 | 1.35 ± .01 | 0.61 ± .00 | 0.72 ± .01 | **0.59** ± .01 | 1.32 ± .00 | **0.06** ± .00 | **2.5** ± .00 | 1.00 ± .00 |
| Peer = 5 | **1.33** ± .04 | **0.60** ± .00 | **0.71** ± .01 | 0.60 ± .00 | 1.32 ± .01 | **0.06** ± .00 | **2.5** ± .00 | **0.99**± .00 |
| Peer = 10 | **1.33** ± .01 | 0.61 ± .00 | **0.71** ± .00 | 0.60 ± .00 | **1.31** ± .01 | 0.07 ± .00 | **2.5** ± .00 | 1.00 ± .00 |
| Peer = 15 | 1.34 ± .00 | 0.61 ± .00 | 0.72 ± .01 | 0.60 ± .00 | **1.31** ± .00 | 0.07 ± .00 | **2.5** ± .00 | **0.99** ± .00 |
| Peer = 20 | 1.34 ± .04 | 0.61 ± .00 | 0.72 ± .01 | 0.60 ± .01 | **1.31** ± .00 | 0.07 ± .00 | **2.5** ± .00 | 1.00 ± .00 |
| $\epsilon = 0$ | 1.34 ± .01 | 0.61 ± .00 | **0.71** ± .01 | 0.60 ± .01 | **1.30** ± .00 | **0.06** ± .00 | **2.50** ± .00 | 0.99 ± .00 |
| $\epsilon = 10^{-5}$ | **1.31** ± .01 | 0.61 ± .00 | **0.71** ± .00 | 0.60 ± .01 | **1.30** ± .00 | 0.07 ± .00 | **2.50** ± .00 | 1.00 ± .00 |
| $\epsilon = 10^{-4}$ | 1.33 ± .04 | **0.60** ± .00 | **0.71** ± .01 | 0.60 ± .00 | **1.30** ± .00 | **0.06** ± .00 | **2.50** ± .00 | **0.99** ± .00 |
| $\epsilon = 10^{-3}$ | 1.33 ± .04 | **0.60** ± .00 | 0.72 ± .01 | 0.60 ± .01 | **1.30** ± .00 | 0.07 ± .01 | **2.50** ± .00 | **0.99** ± .00 |

**Initialization parameter $\epsilon$:** We also evaluate whether a non-zero value of $\epsilon$ in the initialization process of M³-Impute indeed lead to an improvement in imputation accuracy. As shown in Table 3, for YACHT and WINE datasets, the introduction of a non-zero value of $\epsilon$ results in lower MAE scores. Another insight that we have from Table 3 is that $\epsilon$ should not be set too large, as a large value of $\epsilon$ might impose incorrect weights to the features with missing values. We observe that it is an overall good choice to set $\epsilon$ to $1 \times 10^{-5}$ or $1 \times 10^{-4}$.

## 5 Conclusion

We have presented M³-Impute, a mask-guided representation learning for missing data imputation. M³-Impute improved the initialization process by considering the relationships between samples and their associated features (including missing ones) even in initializing the embeddings. In addition, for more effective representation learning, we introduced two novel components in M³-Impute – **FCU** and **SCU**, which learn feature-wise and sample-wise correlations, respectively, to capture data correlations explicitly and leverage them for imputation. Extensive experiment results demonstrate the effectiveness of M³-Impute. M³-Impute achieves overall superior performance to popular and state-of-the-art methods on 15 open datasets, with 13 best and two second-best MAE scores on average under three different settings of missing value patterns.

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

# A  Appendix

Table 4: Overview of Datasets.

|  | Concrete | Housing | Wine | Yacht | Energy | Kin8nm | Naval | Power |
|---|---|---|---|---|---|---|---|---|
| # Samples | 1030 | 506 | 1599 | 308 | 768 | 8192 | 11934 | 9568 |
| # Features | 8 | 13 | 11 | 6 | 8 | 8 | 16 | 4 |

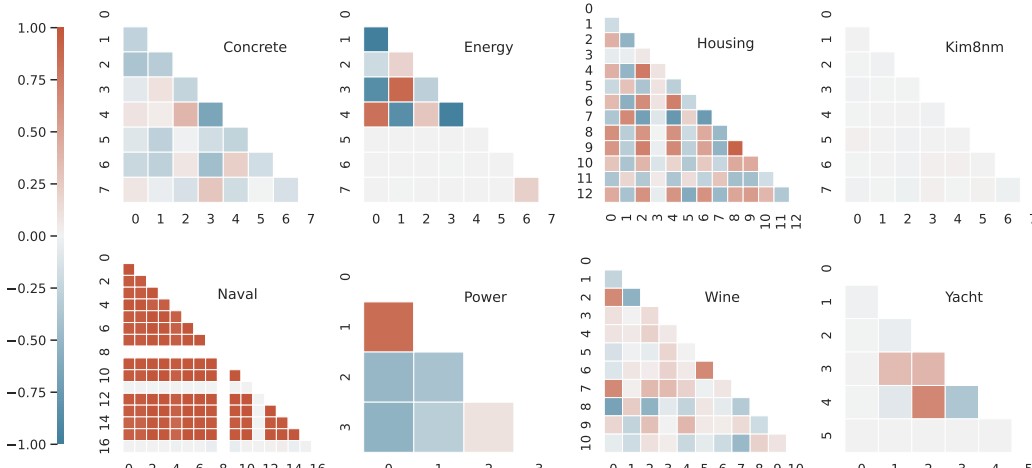

Figure 4: Pearson correlation coefficients of UCI datasets.

In this section, we discuss further experimental details. We first give an overview of the dataset details in Section A.1, followed by the implementation of different missing types and present corresponding imputation performance under MAR and MNAR settings (Section A.2). We then provide the comprehensive results of the robustness experiments (Section A.3). Finally, we extend our evaluation of $M^3$-Impute to seven additional datasets (Section A.4) and elaborate on the computational resources in Section A.5.

## A.1  Dataset Details

Table 4 presents the statistics of the eight UCI datasets [11] used throughout Section 4. Figure 4 illustrates the Pearson correlation coefficients among the features. In the Kin8nm dataset, all features are linearly independent, whereas the Naval dataset exhibits strong correlations among its features. Under the MCAR setting, $M^3$-Impute performs comparably to the baseline imputation methods on these two datasets (shown in Table 1). However, in real-world scenarios, features are not always entirely independent or strongly correlated. In the other six datasets, we observe a mix of weakly correlated features along with a few that are strongly correlated. In these cases, $M^3$-Impute consistently outperforms all baseline methods.

## A.2  Detailed Results of Different Missing Types

We adopt the same procedure outlined in [52, 54] to generate missing values under different settings.

- **MCAR**: A $n \times m$ matrix is sampled from a uniform distribution. Positions with values no greater than the ratio of missingness are viewed as missing and the remaining positions are observable.
- **MAR**: First, a subset of features is randomly selected to be fully observed. Then, these remaining features have values removed according to a logistic model with random weights, using the fully observed feature values as input. The desired rate of missingness is achieved by adjusting the bias term.
- **MNAR**: This is done by first apply the MAR mechanism above. Then, the remaining feature values are masked out by the MCAR mechanism.

Table 5: MAE scores under MAR setting.

| | Yacht | Wine | Concrete | Housing | Energy | Naval | Kin8nm | Power |
|---|---|---|---|---|---|---|---|---|
| Mean | 2.20 | 1.09 | 1.79 | 2.02 | 3.26 | 2.75 | **2.49** | 1.81 |
| Svd [18] | 2.64 | 1.04 | 2.32 | 1.71 | 3.68 | 0.52 | 2.69 | 2.37 |
| Spectral [30] | 3.06 | 0.91 | 2.12 | 1.84 | 2.88 | 1.29 | 3.56 | 3.37 |
| Mice [45] | 1.79 | 0.79 | 1.27 | 1.22 | 1.12 | 0.27 | 2.51 | 1.16 |
| Knn [43] | 1.69 | 0.66 | 0.89 | 0.89 | 1.61 | **0.07** | 2.94 | 1.11 |
| Gain [50] | 2.07 | 1.13 | 1.87 | 0.92 | 2.26 | 0.91 | 2.93 | 1.42 |
| Miwae [29] | 3.47 | 1.04 | 1.87 | 3.79 | 3.82 | 3.78 | 2.57 | 2.07 |
| Grape [52] | 1.20 | **0.60** | **0.77** | 0.66 | 1.05 | **0.07** | **2.49** | 1.06 |
| Miracle [24] | 44.33 | 1.70 | 3.08 | 48.63 | 38.20 | 48.77 | 2.82 | 0.86 |
| HyperImpute [23] | 2.06 | 0.78 | 1.30 | 1.05 | 1.11 | 1.01 | 3.07 | 1.07 |
| M$^3$-Impute | **1.09** | **0.60** | **0.77** | **0.60** | **0.98** | **0.07** | **2.49** | **1.01** |

Table 6: MAE scores under MNAR setting.

| | Yacht | Wine | Concrete | Housing | Energy | Naval | Kin8nm | Power |
|---|---|---|---|---|---|---|---|---|
| Mean | 2.18 | 1.04 | 1.80 | 1.95 | 3.17 | 2.60 | 2.49 | 1.76 |
| Svd [18] | 2.61 | 1.06 | 2.24 | 1.58 | 3.55 | 0.53 | 2.69 | 2.27 |
| Spectral [30] | 2.75 | 1.01 | 1.86 | 1.60 | 2.50 | 1.35 | 3.34 | 3.14 |
| Mice [45] | 1.91 | 0.77 | 1.37 | 1.22 | 1.57 | 0.21 | 2.50 | 1.08 |
| Knn [43] | 1.92 | 0.75 | 1.15 | 0.95 | 1.96 | **0.08** | 3.06 | 1.65 |
| Gain [50] | 2.34 | 0.92 | 1.80 | 1.08 | 1.92 | 1.12 | 2.78 | 1.22 |
| Miwae [29] | 3.77 | 1.02 | 1.86 | 3.80 | 2.74 | 3.79 | 2.58 | 1.93 |
| Grape [52] | 1.23 | 0.61 | 0.73 | 0.61 | 1.16 | **0.08** | **2.46** | 1.02 |
| Miracle [24] | 43.57 | 1.03 | 2.15 | 46.17 | 39.37 | 46.50 | 2.64 | 1.06 |
| HyperImpute [23] | 1.95 | 0.72 | 0.88 | 0.85 | 1.19 | 0.85 | 2.71 | 1.09 |
| M$^3$-Impute | **1.15** | **0.60** | **0.68** | **0.54** | **1.09** | **0.08** | **2.46** | **1.00** |

In addition to the results for MCAR setting presented in Table 4.2, Table 5 and Table 6 present the MAE scores under MAR and MNAR settings, respectively. M$^3$-Impute consistently outperforms all baseline methods in both scenarios.

## A.3 Robustness against Various Ratios of Missingness

Table 8 presents the performance of various imputation methods across different ratios of missingness. M$^3$-Impute achieves the lowest MAE scores in most cases and the second-best MAE scores in the remaining ones.

## A.4 Further Evaluation on Seven Additional Datasets

Table 7: Overview of seven additional datasets.

| | airfoil | blood | wine-white | ionosphere | breast | iris | diabetes |
|---|---|---|---|---|---|---|---|
| # Samples | 1503 | 748 | 4899 | 351 | 569 | 150 | 442 |
| # Features | 6 | 4 | 12 | 34 | 30 | 4 | 10 |

In this experiment, we further evaluate M$^3$-Impute on seven datasets: Airfoil [3], Blood [49], Wine-White [7], Ionosphere [39], Breast Cancer [47], Iris [13], and Diabetes [12]. An overview of dataset details is provided in Table 7, and feature correlations are illustrated in Figure 5. We simulate missingness in data under MCAR, MAR, and MNAR conditions, each with a missing ratio of 0.3. Results are demonstrated in Table 9. Across all three types of missingness, M$^3$-Impute achieves five best and two second-best MAE scores on average.

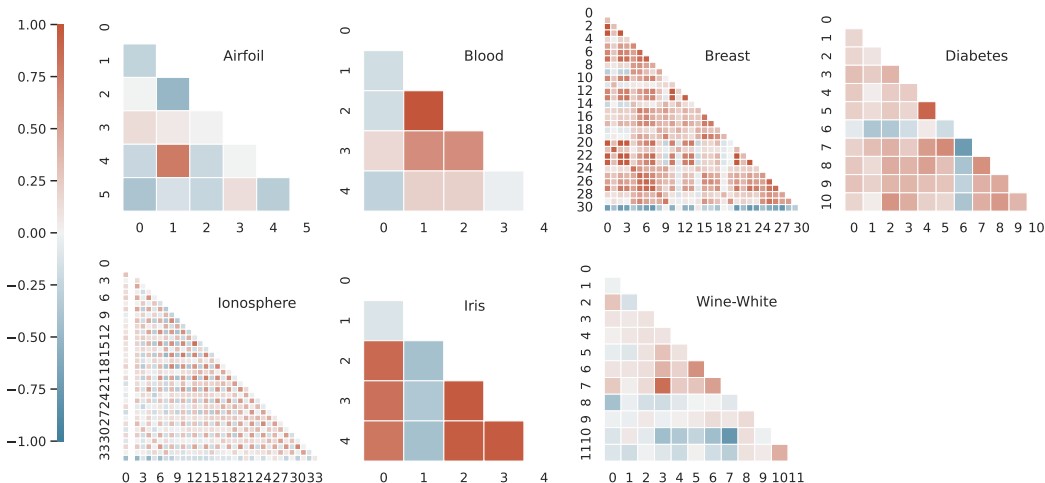

Figure 5: Pearson correlation coefficient of 7 extra datasets.

## A.5 Computational Resources

All our experiments are conducted on a GPU server running Ubuntu 22.04, with PyTorch 2.1.0 and CUDA 12.1. We train and test M$^3$-Impute using a single NVIDIA A100 80G GPU. With the experimental setup described in Section 4.1, the total runtime (including both training and testing) for each of the five repeated runs ranged from 1 to 5 hours, depending on the scale of the datasets.

Table 8: MAE scores across different levels of missingness.

| | Yacht | Wine | Concrete | Housing | Energy | Naval | Kin8nm | Power |
|---|---|---|---|---|---|---|---|---|
| **Missing 10%** | | | | | | | | |
| Mean | 2.22 ± 0.05 | 0.96 ± 0.02 | 1.81 ± 0.02 | 1.84 ± 0.01 | 3.09 ± 0.07 | 2.30 ± 0.01 | 2.50 ± 0.01 | 1.68 ± 0.00 |
| Svd | 1.92 ± 0.16 | 0.88 ± 0.03 | 2.04 ± 0.04 | 1.69 ± 0.11 | 1.75 ± 0.10 | 0.34 ± 0.00 | 5.04 ± 0.06 | 2.26 ± 0.04 |
| Spectral | 2.24 ± 0.12 | 0.76 ± 0.02 | 1.84 ± 0.05 | 1.28 ± 0.04 | 1.76 ± 0.08 | 0.38 ± 0.01 | 2.71 ± 0.02 | 1.77 ± 0.02 |
| Mice | 1.38 ± 0.13 | 0.62 ± 0.01 | 0.97 ± 0.04 | 0.98 ± 0.04 | 1.28 ± 0.07 | 0.13 ± 0.00 | 2.50 ± 0.01 | 1.01 ± 0.01 |
| Knn | 1.40 ± 0.17 | 0.49 ± 0.01 | 0.58 ± 0.05 | 0.74 ± 0.04 | 1.42 ± 0.05 | **0.03** ± 0.00 | 2.53 ± 0.01 | 1.26 ± 0.00 |
| Gain | 2.30 ± 0.04 | 0.83 ± 0.04 | 1.62 ± 0.05 | 1.16 ± 0.05 | 1.95 ± 0.05 | 0.45 ± 0.01 | 2.74 ± 0.02 | 1.22 ± 0.00 |
| Miwae | 4.57 ± 0.09 | 0.98 ± 0.01 | 1.85 ± 0.03 | 3.78 ± 0.10 | 2.77 ± 0.16 | 2.36 ± 0.00 | 2.56 ± 0.00 | 1.74 ± 0.00 |
| Grape | 1.00 ± 0.00 | 0.48 ± 0.00 | 0.45 ± 0.01 | 0.49 ± 0.00 | 1.19 ± 0.00 | 0.05 ± 0.00 | 2.49 ± 0.00 | 0.85 ± 0.03 |
| Miracle | 44.77 ± 0.05 | 0.97 ± 0.19 | 1.91 ± 0.07 | 43.90 ± 0.33 | 41.43 ± 0.34 | 0.12 ± 0.00 | **2.48** ± 0.00 | 1.07 ± 0.05 |
| HyperImpute | 1.50 ± 0.11 | 0.52 ± 0.00 | 0.51 ± 0.04 | 0.75 ± 0.04 | 1.18 ± 0.05 | 0.06 ± 0.04 | 2.50 ± 0.00 | **0.84** ± 0.00 |
| M³-Impute | **0.96** ± 0.00 | **0.47** ± 0.01 | **0.41** ± 0.01 | **0.45** ± 0.00 | **1.15** ± 0.00 | 0.05 ± 0.00 | 2.49 ± 0.00 | **0.84** ± 0.01 |
| | Yacht | Wine | Concrete | Housing | Energy | Naval | Kin8nm | Power |
| **Missing 30%** | | | | | | | | |
| Mean | 2.09 ± 0.04 | 0.98 ± 0.01 | 1.79 ± 0.01 | 1.85 ± 0.00 | 3.10 ± 0.04 | 2.31 ± 0.00 | 2.50 ± 0.00 | 1.68 ± 0.00 |
| Svd | 2.46 ± 0.16 | 0.92 ± 0.01 | 1.94 ± 0.02 | 1.53 ± 0.03 | 2.24 ± 0.06 | 0.50 ± 0.00 | 3.67 ± 0.06 | 2.33 ± 0.01 |
| Spectral | 2.64 ± 0.11 | 0.91 ± 0.01 | 1.98 ± 0.04 | 1.46 ± 0.03 | 2.26 ± 0.09 | 0.41 ± 0.00 | 2.80 ± 0.01 | 2.13 ± 0.01 |
| Mice | 1.68 ± 0.05 | 0.77 ± 0.00 | 1.34 ± 0.01 | 1.16 ± 0.03 | 1.53 ± 0.04 | 0.20 ± 0.01 | 2.50 ± 0.00 | 1.16 ± 0.01 |
| Knn | 1.67 ± 0.02 | 0.72 ± 0.00 | 1.16 ± 0.03 | 0.95 ± 0.01 | 1.81 ± 0.03 | 0.10 ± 0.00 | 2.77 ± 0.01 | 1.38 ± 0.01 |
| Gain | 2.26 ± 0.11 | 0.86 ± 0.00 | 1.67 ± 0.03 | 1.23 ± 0.02 | 1.99 ± 0.03 | 0.46 ± 0.02 | 2.70 ± 0.00 | 1.31 ± 0.05 |
| Miwae | 4.68 ± 0.16 | 1.00 ± 0.00 | 1.81 ± 0.01 | 3.81 ± 0.04 | 2.79 ± 0.04 | 2.37 ± 0.00 | 2.57 ± 0.00 | 1.74 ± 0.00 |
| Grape | 1.46 ± 0.01 | 0.60 ± 0.00 | 0.75 ± 0.01 | 0.64 ± 0.01 | 1.36 ± 0.01 | 0.07 ± 0.00 | 2.50 ± 0.00 | 1.00 ± 0.00 |
| Miracle | 42.97 ± 0.53 | 1.13 ± 0.00 | 1.71 ± 0.05 | 42.23 ± 0.31 | 41.43 ± 0.34 | 0.17 ± 0.00 | 2.49 ± 0.00 | 1.15 ± 0.01 |
| HyperImpute | 1.76 ± 0.03 | 0.67 ± 0.01 | 0.84 ± 0.02 | 0.82 ± 0.01 | **1.32** ± 0.02 | **0.04** ± 0.00 | 2.58 ± 0.05 | 1.06 ± 0.01 |
| M³-Impute | **1.33** ± 0.04 | **0.60** ± 0.00 | **0.71** ± 0.01 | **0.60** ± 0.00 | 1.32 ± 0.01 | 0.06 ± 0.00 | 2.50 ± 0.00 | **0.99** ± 0.00 |
| | Yacht | Wine | Concrete | Housing | Energy | Naval | Kin8nm | Power |
| **Missing 50%** | | | | | | | | |
| Mean | 2.12 ± 0.02 | 0.98 ± 0.01 | 1.81 ± 0.01 | 1.84 ± 0.01 | 3.08 ± 0.02 | 2.31 ± 0.00 | **2.50** ± 0.00 | 1.67 ± 0.00 |
| Svd | 3.00 ± 0.11 | 1.18 ± 0.00 | 2.19 ± 0.01 | 1.88 ± 0.01 | 2.88 ± 0.04 | 0.87 ± 0.00 | 3.30 ± 0.01 | 2.92 ± 0.02 |
| Spectral | 3.17 ± 0.13 | 1.13 ± 0.00 | 2.31 ± 0.01 | 1.76 ± 0.03 | 3.03 ± 0.02 | 0.46 ± 0.00 | 3.02 ± 0.00 | 2.98 ± 0.02 |
| Mice | 1.99 ± 0.08 | 0.83 ± 0.00 | 1.59 ± 0.03 | 1.33 ± 0.02 | 2.13 ± 0.12 | 0.31 ± 0.01 | 2.50 ± 0.00 | 1.32 ± 0.01 |
| Knn | 2.08 ± 0.02 | 0.98 ± 0.01 | 1.40 ± 0.02 | 1.37 ± 0.01 | 2.21 ± 0.01 | 0.76 ± 0.01 | 2.65 ± 0.00 | 1.80 ± 0.01 |
| Gain | 2.33 ± 0.03 | 1.18 ± 0.15 | 2.20 ± 0.17 | 1.43 ± 0.09 | 2.58 ± 0.09 | 0.56 ± 0.03 | 2.86 ± 0.06 | 1.36 ± 0.00 |
| Miwae | 4.57 ± 0.06 | 1.01 ± 0.01 | 1.85 ± 0.02 | 3.79 ± 0.01 | 2.83 ± 0.05 | 2.38 ± 0.00 | 2.58 ± 0.00 | 1.73 ± 0.00 |
| Grape | 1.89 ± 0.02 | 0.75 ± 0.01 | 1.24 ± 0.00 | 0.83 ± 0.01 | 1.63 ± 0.01 | 0.09 ± 0.00 | 2.50 ± 0.00 | **1.19** ± 0.00 |
| Miracle | 40.77 ± 0.34 | 1.08 ± 0.00 | 2.00 ± 0.08 | 39.40 ± 0.33 | 37.40 ± 0.22 | 0.24 ± 0.00 | 2.82 ± 0.06 | 1.29 ± 0.00 |
| HyperImpute | 2.07 ± 0.11 | 0.85 ± 0.00 | 1.33 ± 0.08 | 1.06 ± 0.11 | 1.70 ± 0.05 | **0.07** ± 0.00 | 2.96 ± 0.04 | 1.29 ± 0.01 |
| M³-Impute | **1.74** ± 0.01 | **0.74** ± 0.00 | **1.19** ± 0.02 | **0.79** ± 0.01 | **1.57** ± 0.00 | 0.08 ± 0.00 | 2.50 ± 0.00 | **1.19** ± 0.00 |
| | Yacht | Wine | Concrete | Housing | Energy | Naval | Kin8nm | Power |
| **Missing 70%** | | | | | | | | |
| Mean | 2.16 ± 0.06 | 0.99 ± 0.00 | 1.81 ± 0.01 | 1.83 ± 0.02 | 3.08 ± 0.01 | 2.31 ± 0.00 | 2.50 ± 0.00 | 1.67 ± 0.00 |
| Svd | 3.78 ± 0.06 | 1.63 ± 0.02 | 2.53 ± 0.03 | 2.58 ± 0.07 | 3.65 ± 0.09 | 1.56 ± 0.00 | 3.58 ± 0.00 | 3.88 ± 0.01 |
| Spectral | 4.17 ± 0.10 | 1.67 ± 0.02 | 2.75 ± 0.01 | 2.59 ± 0.05 | 4.00 ± 0.03 | 1.04 ± 0.00 | 3.73 ± 0.01 | 4.33 ± 0.01 |
| Mice | 2.21 ± 0.10 | 0.93 ± 0.01 | 1.72 ± 0.02 | 1.54 ± 0.04 | 2.71 ± 0.15 | 0.53 ± 0.00 | 2.62 ± 0.08 | 1.46 ± 0.00 |
| Knn | 2.62 ± 0.08 | 1.05 ± 0.00 | 1.60 ± 0.01 | 1.43 ± 0.02 | 2.54 ± 0.04 | 1.08 ± 0.00 | 2.84 ± 0.01 | 2.73 ± 0.00 |
| Gain | 3.07 ± 0.08 | 1.61 ± 0.15 | 2.84 ± 0.04 | 3.09 ± 0.04 | 3.83 ± 0.15 | 1.07 ± 0.02 | 3.31 ± 0.21 | 1.51 ± 0.05 |
| Miwae | 4.56 ± 0.07 | 1.02 ± 0.00 | 1.84 ± 0.01 | 3.78 ± 0.02 | 3.02 ± 0.07 | 2.38 ± 0.00 | 2.58 ± 0.00 | 1.72 ± 0.00 |
| Grape | **2.14** ± 0.01 | 0.88 ± 0.01 | 1.64 ± 0.02 | 1.12 ± 0.01 | 2.10 ± 0.01 | 0.17 ± 0.00 | **2.49** ± 0.00 | 1.37 ± 0.00 |
| Miracle | 38.37 ± 0.38 | 1.03 ± 0.00 | 2.45 ± 0.21 | 36.23 ± 0.21 | 33.93 ± 0.17 | 0.53 ± 0.00 | 3.09 ± 0.02 | 1.92 ± 0.04 |
| HyperImpute | 2.49 ± 0.08 | 0.92 ± 0.02 | 1.71 ± 0.01 | 1.12 ± 0.13 | 2.16 ± 0.06 | **0.15** ± 0.00 | 3.15 ± 0.03 | 1.54 ± 0.02 |
| M³-Impute | **2.14** ± 0.00 | **0.87** ± 0.00 | **1.56** ± 0.01 | **1.08** ± 0.00 | **2.05** ± 0.00 | 0.17 ± 0.00 | **2.49** ± 0.00 | **1.37** ± 0.00 |

Table 9: MAE scores on seven additional datasets

| | airfoil | blood | wine-white | ionosphere | breast | iris | diabetes |
|---|---|---|---|---|---|---|---|
| **MCAR** | | | | | | | |
| Mean | 2.32 ± 0.05 | 1.14 ± 0.01 | 0.76 ± 0.00 | 2.01 ± 0.03 | 1.06 ± 0.00 | 2.15 ± 0.09 | 1.78 ± 0.03 |
| Svd | 2.76 ± 0.05 | 0.97 ± 0.04 | 0.87 ± 0.00 | 1.26 ± 0.03 | 0.58 ± 0.00 | 1.70 ± 0.07 | 1.76 ± 0.02 |
| Spectral | 2.30 ± 0.07 | 0.94 ± 0.03 | 0.78 ± 0.01 | 1.38 ± 0.02 | 0.38 ± 0.00 | 1.48 ± 0.13 | 1.48 ± 0.03 |
| Mice | 1.97 ± 0.04 | 0.69 ± 0.01 | 0.61 ± 0.01 | 1.37 ± 0.03 | 0.34 ± 0.01 | 1.07 ± 0.09 | 1.29 ± 0.05 |
| Knn | 2.18 ± 0.04 | 0.93 ± 0.01 | 0.64 ± 0.01 | 1.07 ± 0.03 | 0.53 ± 0.01 | 1.54 ± 0.22 | 1.71 ± 0.04 |
| Gain | 2.22 ± 0.06 | 1.26 ± 0.04 | 0.73 ± 0.01 | 1.50 ± 0.01 | 0.51 ± 0.01 | 1.29 ± 0.07 | 1.47 ± 0.06 |
| Miracle | 2.13 ± 0.05 | 43.17 ± 0.05 | 0.60 ± 0.00 | 37.70 ± 0.22 | 35.07 ± 0.41 | 45.13 ± 0.42 | 41.00 ± 0.14 |
| Grape | 1.16 ± 0.02 | 0.68 ± 0.00 | **0.52** ± 0.00 | 1.08 ± 0.01 | 0.37 ± 0.00 | **0.82** ± 0.00 | 1.31 ± 0.00 |
| Miwae | 2.36 ± 0.06 | 2.03 ± 0.05 | 0.77 ± 0.00 | 5.14 ± 0.06 | 1.89 ± 0.02 | 4.60 ± 0.17 | 5.05 ± 0.04 |
| HyperImpute | **1.09** ± 0.02 | **0.63** ± 0.02 | 0.55 ± 0.00 | 1.18 ± 0.04 | **0.33** ± 0.01 | 1.04 ± 0.11 | **1.17** ± 0.02 |
| M³-Impute | **1.09** ± 0.03 | 0.67 ± 0.00 | **0.52** ± 0.00 | **1.01** ± 0.01 | 0.36 ± 0.01 | **0.82** ± 0.00 | 1.29 ± 0.01 |
| **MAR** | | | | | | | |
| Mean | 2.33 ± 0.14 | 0.91 ± 0.02 | 0.87 ± 0.01 | 2.02 ± 0.08 | 1.13 ± 0.03 | 1.99 ± 0.25 | 1.74 ± 0.33 |
| Svd | 2.99 ± 0.83 | 0.91 ± 0.07 | 0.78 ± 0.05 | 1.40 ± 0.08 | 0.61 ± 0.03 | 1.85 ± 0.42 | 2.09 ± 0.02 |
| Spectral | 2.01 ± 0.60 | 1.22 ± 0.36 | 0.99 ± 0.23 | 1.50 ± 0.02 | 0.46 ± 0.04 | 1.62 ± 0.13 | 1.32 ± 0.20 |
| Mice | 2.16 ± 0.28 | 1.00 ± 0.40 | 0.63 ± 0.04 | 1.43 ± 0.08 | 0.32 ± 0.07 | 0.85 ± 0.09 | 1.33 ± 0.23 |
| Knn | 1.59 ± 0.70 | 0.90 ± 0.25 | 0.53 ± 0.02 | 1.09 ± 0.03 | 0.53 ± 0.03 | 0.91 ± 0.08 | 1.43 ± 0.23 |
| Gain | 2.29 ± 0.09 | 1.01 ± 0.15 | 0.65 ± 0.11 | 1.71 ± 0.10 | 0.69 ± 0.05 | 1.25 ± 0.04 | 1.34 ± 0.04 |
| Miracle | 2.08 ± 0.26 | 42.30 ± 0.22 | 1.05 ± 0.05 | 26.60 ± 0.37 | 39.53 ± 0.17 | 49.60 ± 1.14 | 41.83 ± 0.09 |
| Grape | 1.57 ± 0.02 | 0.29 ± 0.01 | **0.48** ± 0.00 | 1.17 ± 0.03 | 0.39 ± 0.00 | 0.86 ± 0.02 | 1.12 ± 0.01 |
| Miwae | 2.56 ± 0.01 | 2.03 ± 0.03 | 0.69 ± 0.01 | 6.10 ± 0.04 | 2.17 ± 0.03 | 3.46 ± 0.13 | 4.26 ± 0.06 |
| HyperImpute | **1.21** ± 0.21 | 0.88 ± 0.33 | 0.57 ± 0.08 | 1.30 ± 0.03 | **0.34** ± 0.02 | 1.05 ± 0.11 | 1.46 ± 0.10 |
| M³-Impute | 1.54 ± 0.02 | **0.28** ± 0.01 | **0.48** ± 0.00 | **1.07** ± 0.01 | 0.37 ± 0.01 | **0.82** ± 0.03 | **1.07** ± 0.00 |
| **MNAR** | | | | | | | |
| Mean | 2.36 ± 0.11 | 0.98 ± 0.05 | 0.82 ± 0.01 | 2.04 ± 0.06 | 1.11 ± 0.02 | 2.06 ± 0.09 | 1.77 ± 0.20 |
| Svd | 2.98 ± 0.52 | 0.98 ± 0.09 | 0.82 ± 0.04 | 1.36 ± 0.07 | 0.60 ± 0.03 | 1.66 ± 0.20 | 1.93 ± 0.02 |
| Spectral | 2.64 ± 0.18 | 1.40 ± 0.18 | 0.88 ± 0.13 | 1.46 ± 0.02 | 0.41 ± 0.03 | 1.35 ± 0.11 | 1.51 ± 0.13 |
| Mice | 2.07 ± 0.14 | 0.76 ± 0.17 | 0.62 ± 0.02 | 1.44 ± 0.07 | 0.33 ± 0.02 | 0.99 ± 0.11 | 1.27 ± 0.16 |
| Knn | 2.11 ± 0.27 | 1.04 ± 0.12 | 0.60 ± 0.02 | 1.12 ± 0.03 | 0.55 ± 0.02 | 1.53 ± 0.52 | 1.60 ± 0.17 |
| Gain | 2.21 ± 0.05 | 1.09 ± 0.06 | 0.69 ± 0.01 | 1.55 ± 0.03 | 0.62 ± 0.02 | 1.26 ± 0.04 | 1.43 ± 0.06 |
| Miracle | 1.72 ± 0.08 | 42.90 ± 0.14 | 0.59 ± 0.01 | 30.70 ± 0.57 | 37.30 ± 0.29 | 47.37 ± 0.90 | 41.60 ± 0.37 |
| Grape | 1.46 ± 0.03 | 0.42 ± 0.00 | **0.49** ± 0.00 | 1.15 ± 0.01 | 0.38 ± 0.00 | 0.89 ± 0.02 | 1.21 ± 0.01 |
| Miwae | 2.47 ± 0.03 | 1.99 ± 0.04 | 0.72 ± 0.00 | 5.66 ± 0.02 | 2.05 ± 0.00 | 3.98 ± 0.32 | 4.62 ± 0.08 |
| HyperImpute | **1.23** ± 0.04 | 0.82 ± 0.18 | 0.58 ± 0.05 | 1.28 ± 0.02 | **0.36** ± 0.03 | 1.07 ± 0.07 | 1.30 ± 0.19 |
| M³-Impute | 1.46 ± 0.01 | **0.41** ± 0.00 | **0.49** ± 0.00 | **1.06** ± 0.02 | **0.36** ± 0.01 | **0.87** ± 0.00 | **1.19** ± 0.00 |

