# OpenReview forum: "M$^3$-Impute: Mask-guided Representation Learning for Missing Value Imputation"
_NeurIPS.cc/2024/Conference — Submitted to NeurIPS 2024_

### Official Review · Reviewer_WjjR · 2024-06-30

**Soundness:** 3
**Presentation:** 2
**Contribution:** 3
**Rating:** 6
**Confidence:** 3

**Summary:**

The paper introduces M3-Impute, a mask-guided representation learning method for missing value imputation. The core idea of M3-Impute is to leverage missingness information as an explicit input to the model through masking schemes. This approach allows M3-Impute to effectively learn both feature-wise and sample-wise correlations, accommodating various types of data missingness. The model employs a variant of GraphSAGE for graph representation learning, incorporating edge embeddings via neighborhood aggregation. It outperforms traditional tabular data models in various benchmark datasets.

**Strengths:**

1. The paper presents a novel approach to missing value imputation through the introduction of a mask-guided representation learning method (M3-Impute). The originality of the work lies in its utilization of missingness information as a model input, employing innovative masking schemes. This allows M3-Impute to accurately capture feature-wise and sample-wise correlations despite varying types of missing data (MCAR, MAR, MNAR). The use of GraphSAGE for graph representation learning, combined with edge embeddings via neighborhood aggregation, further distinguishes this work from traditional tabular data models.

2. The quality of the research is demonstrated through comprehensive experiments across multiple datasets and missing data mechanisms. The empirical results show that M3-Impute consistently outperforms baseline methods. The authors include a code package and datasets with the submission.

**Weaknesses:**

1. The paper evaluates the sensitivity of the M3-Impute model to the initialization parameter ϵ (Table 3), demonstrating that a non-zero value of ϵ improves imputation accuracy. However, the lack of detailed sensitivity analysis for other critical hyperparameters, such as the learning rate, batch size, number of GNN layers, and the dropout rate, represents a weakness.

2. The paper also has notable limitations in its contextualization relative to prior work. While it effectively presents M3-Impute and compares it against several baseline models, it lacks a deeper analysis of how these baseline models have evolved and the specific innovations they have introduced over time. For instance, the paper mentions GRAPE and IGRM as key prior graph-based imputation methods but does not adequately explore their strengths and weaknesses or how M3-Impute directly addresses the limitations of these methods. This omission makes it challenging to understand the novelty and improvements offered by M3-Impute.

3. Relying solely on MAE to evaluate the performance of imputation models has several limitations. MAE measures the average magnitude of errors but does not account for the variance or distribution of those errors, making it insensitive to outliers and providing no insight into model bias. This can result in an incomplete understanding of a model's performance, particularly in contexts where large errors or systematic biases are important considerations. To address these limitations, incorporating RMSE alongside MAE would be beneficial. RMSE penalizes larger errors more heavily, offering additional insight into the presence and impact of significant errors in the model's predictions.

**Questions:**

- The statement that most learning-based methods are``built upon the raw tabular data structure as is, which greatly restricts them from jointly modeling the feature-wise and sample-wise correlations” (line 41) is not entirely accurate for two prominent tabular generative models, MIDA and GAIN. MIDA transforms raw tabular data into a higher-dimensional space through its encoder-decoder architecture. This transformation allows MIDA to capture more complex, nonlinear relationships that are not immediately apparent in the raw data. The adversarial process of GAIN allows it to model the joint distribution of the data, thus capturing complex correlations between features and samples.

      - Gondara, L., Wang, K. (2018). MIDA: Multiple Imputation Using Denoising Autoencoders. In: Phung, D., Tseng, V., Webb, G., Ho, B., Ganji, M., Rashidi, L. (eds) Advances in Knowledge Discovery and Data Mining. PAKDD 2018. Lecture Notes in Computer Science(), vol 10939. Springer, Cham. https://doi.org/10.1007/978-3-319-93040-4_21

      - Wang, Zhenhua, et al. "Are deep learning models superior for missing data imputation in surveys? Evidence from an empirical comparison." Survey Methodology 48 (2022): 375-399.

- When discussing statistical methods (line 70), the authors should mention that FCS approaches such as MICE are flexible in imputing different types of variables.

- Related to the previous point, MICE is generally considered a statistical method rather than a learning-based method, although a learning algorithm such as CART can be used as the imputer. See the paper by Wang et. al. (2022).

- In Sec. 4.2, it should be noted that the M3-Impute model tends to perform slightly better under MAR and MNAR settings for most datasets, indicating its effectiveness in handling missingness that depends on the observed data.

- How can interpretability techniques be incorporated into M3-Impute to help users understand the imputation decisions?

**Limitations:**

The authors should consider summarizing the limitations of their method in the conclusion to provide a comprehensive overview of their work. Specifically, in Section 4.2, the authors discuss the cases of MAE degradation for the Kin8nm and Naval datasets. They attribute this to the independence of features in Kin8nm, which prevents observed features from aiding in the imputation of missing values, and the strong linear correlations between nearly all features in the Naval dataset. Summarizing these points in the conclusion would give readers a clear understanding of the method's limitations and the contexts in which it performs best.

---

> ### Author Rebuttal · Authors · 2024-08-07
>
> We thank the reviewer for recognizing our method as a novel approach with innovative masking schemes and that our experiments comprehensively demonstrate superior performance of our method to baseline methods. We also appreciate the constructive comments. Below we provide our response to the concerns raised.
>
>  **Q1. Detailed sensitivity analysis for other critical hyperparameters.**
>
> Thanks for the comment. The results of a detailed sensitivity analysis for other critical hyperparameters are presented in Tables 5-7 in the rebuttal PDF. We will incorporate them into our final manuscript.
>
>  **Q2. Differentiation with GRAPE and IGRM.**
>
> Thanks for pointing this out. GRAPE is a pioneering work that models tabular data with missing values as a bipartite graph. IGRM builds on the bipartite graph modeling and further computes sample correlations to connect similar samples by creating new edges for learning. However, they consider each sample or feature as a whole in computing correlations, whereas in M$^3$-Impute, we treat each entry in the tabular dataset individually to compute its correlation with other entries. This is achieved by our novel soft masking design with Feature Correlation Unit (FCU) and Sample Correlation Unit (SCU). We will clarify this point in the final manuscript. Thanks.
>
>  **Q3. RMSE comparison between M$^3$-Impute and baseline methods**
>
> Thanks for the suggestion. The RMSE results are now provided in Tab.1 below, which shows that our method outperforms the baselines as well. We will incorporate them into the final manuscript.
>
> **Tab.1: RMSE under MCAR setting with 30\% missingness. RMSE scores are enlarged by 10 times for better clarity.**
>
> |         | Yacht | Wine | Concrete | Housing | Energy | Naval | Kin8nm | Power |
> |---------|:-------:|:------:|:----------:|:---------:|:--------:|:-------:|:--------:|:-------:|
> | Mean    | 2.92  | 1.33 | 2.23     | 2.49    | 3.54   | 2.89  | **2.88** | 1.98  |
> | Svd     | 3.07  | 1.28 | 2.65     | 2.11    | 4.09   | 1.18  | 4.75    | 2.74  |
> | Spectral| 3.20  | 1.28 | 2.59     | 2.14    | 3.04   | 1.05  | 3.37    | 3.72  |
> | Mice    | 2.49  | 1.05 | 1.78     | 1.77    | 2.28   | 0.54  | **2.88** | 1.47  |
> | kNN     | 2.70  | 1.20 | 2.05     | 1.89    | 3.12   | 0.47  | 3.48    | 2.14  |
> | Gain    | 2.90  | 1.16 | 2.21     | 1.87    | 2.70   | 0.89  | 3.23    | 1.63  |
> | Miwae   | 5.93  | 1.41 | 2.41     | 4.70    | 3.78   | 3.03  | 3.03    | 2.08  |
> | Grape   | _2.34_ | **0.89** | _1.27_    | _1.32_   | _2.24_  | _0.18_ | _2.89_   | _1.32_ |
> | Miracle | 43.86 | 1.56 | 2.17     | 43.61   | 42.21  | 0.60  | _2.89_  | 1.46  |
> | HyperImpute | 2.50  | _0.99_ | 1.47     | 1.62    | 2.52   | 0.19  | 3.04    | 1.43  |
> | **M${^3}$-Impute** | **2.27** | **0.89** | **1.23** | **1.28** | **2.19** | **0.16** | _2.89_ | **1.30** |
> |
>
>  **Q4. Misclassification of MIDA and GAIN.**
>
> Thanks for the correction. We will restructure our introduction as suggested to make our arguments more precise in the final manuscript.
>
>  **Q5. When discussing statistical methods (line 70), the authors should mention that FCS approaches such as MICE are flexible in imputing different types of variables.**
>
> Thanks for the comment. We will restructure the related work section and discuss such approaches correctly in the final manuscript.
>
>  **Q6. More precise statement in Sec. 4.2.**
>
> Thanks for the detailed comment. We will make the statements more precise in the final manuscript.
>
>  **Q7. How can interpretability techniques be incorporated into M3-Impute to help users understand the imputation decisions?**
>
> Thanks for the suggestion. We plan to add a plot in the final manuscript to show the intermediate correlation learning process. This plot will illustrate the dependency on other entries when imputing the value of a missing entry in the table.
>
>  **Q8. Summarize limitations in the conclusion.**
>
> Thanks for the suggestion. We will summarize the limitations of our methods, e.g., presence of the datasets where our method is not the top performer, in the conclusion of the final manuscript.

---

> > ### Comment · Reviewer_WjjR · 2024-08-11
> >
> > The authors have addressed the lack of empirical validation by providing additional sensitivity analyses for hyperparameters (Tables 5-7 in the rebuttal PDF) and RMSE results (Table 1 in the rebuttal) as requested.
> >
> > They acknowledge the need to add a limitations section in the conclusion and summarize the cases where their method underperforms (KIN8NM and NAVAL datasets), and commit to adding this in the final manuscript.
> >
> > The broader empirical validation and commitment to adding a limitations section will strengthen the revised paper.

---

### Official Review · Reviewer_Bqr5 · 2024-07-06

**Soundness:** 3
**Presentation:** 2
**Contribution:** 4
**Rating:** 6
**Confidence:** 3

**Summary:**

This study proposes a missing value imputation method. The proposed method tackles the missing value imputation problem as a link prediction task on the bipartite graph. It represents a data matrix with missing values as a bipartite graph, then uses a graph neural network on the bipartite graph to learn the embeddings of samples and features. Next, a feature correlation unit and a sample correlation unit are employed to obtain feature-wise and sample-wise correlations, which are then fed into an MLP to obtain imputed values.

**Strengths:**

- An advanced and novel approach to missing value imputation. The idea is intuitive.
- Experimental comparison is done with various existing methods, including recent ones.
- Significant performance improvements achieved.

**Weaknesses:**

- The size of the bipartite graph may drastically increase with data size and dimensionality.
- No investigation on various missingness scenarios and missing rates.

**Questions:**

- Computational and space complexity analysis is needed. Can the proposed method be used for a very large training dataset with high dimensionality?
- How does the proposed method perform under lower and higher missing rates?
- The inference phase is unclear. Please elaborate on how the proposed method makes imputations for a query instance containing missing values?
- While the authors reviewed other recent methods that leveraged graph neural networks ([40], [52], [54]), only the method in [52] was compared in the experiments. Why were the methods in [40] and [54] not compared?

**Limitations:**

-

---

> ### Author Rebuttal · Authors · 2024-08-07
>
> We thank the reviewer for recognizing our method as an advanced and novel approach that achieves significant performance improvements, and that our contribution is excellent. We also appreciate the constructive comments. Below we respond to the main concerns raised.
>
>  **Q1. The size of the bipartite graph may drastically increase with data size and dimensionality.**
>
> Yes, the size of the bipartite graph would increase with larger datasets. However, as demonstrated in Table 4 in the rebuttal PDF, the computation cost would not increase drastically as the mini-batch neighborhood aggregation is efficient in generating node embeddings, which use a reasonable amount of computational resources.
>
>  **Q2. No investigation on various missingness scenarios and missing rates.**
>
> Thanks for the comment. We would like to point out that we indeed carried out this study, and the results are available in Tables 5, 6, 8, and 9 in the appendix of the submitted manuscript.
>
>  **Q3. Computational and space complexity analysis is needed. Can the proposed method be used for a very large training dataset with high dimensionality?**
>
> Yes, we have conducted experiments on new large datasets, including Protein, Spam (with 57 features), Letter, and California Housing, which have a large number of samples and features. The results confirm that our method is feasible and efficient in handling such large datasets. The dataset statistics and running times can be found in Table 1 and Table 4 in the rebuttal PDF, respectively. Thanks.
>
>  **Q4. How does the proposed method perform under lower and higher missing rates? The inference phase is unclear. Please elaborate on how the proposed method makes imputations for a query instance containing missing values?**
>
> Thanks for the comment. The results of different missing ratios are available in Table 8 in the submitted manuscript.
>
> In the inference phase, the pipeline for imputing the missing values of a given sample is detailed in Algorithm 1 in the manuscript. For instance, given a sample with 5 features, where 2 features are missing, M$^3$-Impute first introduces new sample node and establishes edges between the sample node and the three non-missing feature nodes, which already exist in the original graph. The sample node embedding is initialized as per Equation (2). The embeddings of feature nodes and the sample node are then updated through the neighborhood aggregation process using the node embeddings learned from training.
>
> Next, these updated node embeddings are processed through Feature Correlation Unit (FCU) and Sample Correlation Unit (SCU) to learn the feature-wise and sample-wise correlations. Suppose we are to impute the first missing feature, denoted by $f$. FCU computes the correlations between $f$ and all three non-missing features, resulting in the vector $c^{f}_s$. In addition, SCU first measures the similarity between the query sample and a set of peer samples when imputing the missing feature $f$ according to Equation (6), and then fuses the information from the peer samples as in Equation (9), resulting in the vector $z^{f}_s$. Finally, the missing feature $f$ is imputed using Equation (11).
>
>  **Q5: While the authors reviewed other recent methods that leveraged graph neural networks ([40], [52], [54]), only the method in [52] was compared in the experiments. Why were the methods in [40] and [54] not compared?**
>
> Thanks for the comment. The code of [54] was unavailable at the submission time of our paper. While their code was recently available, we have not been able to complete running their code on all the 25 datasets without out of memory errors on our experiment platform (NVIDIA A100 GPU with 80GB memory). Hence, we don't report the results here. Nonetheless, we have been able to run the code of [40] and report the results in Tab.1.
>
> ### Tab. 1: Imputation accuracy in MAE under MCAR setting with 30\% missingness
>
> |               | Yacht | Wine | Concrete | Housing | Energy | Naval | Kin8nm | Power |
> |---------------|-------|------|----------|---------|--------|-------|--------|-------|
> | GINN [40]     | 6.89  | 9.49 | 142.85   | 60.66   | 55.06  | 3529.54 | 2.16  | 29.91 |
> | M${^3}$-Impute | **1.33** | **0.60** | **0.71**    | **0.59**   | **1.31**  | **0.06**  | **2.50**  | **0.99** |
> |

---

### Official Review · Reviewer_E1Qa · 2024-07-09

**Soundness:** 2
**Presentation:** 2
**Contribution:** 2
**Rating:** 5
**Confidence:** 3

**Summary:**

This paper addresses the challenge of missing values in data analysis and machine learning by proposing M3-Impute, a novel imputation method. Traditional imputation techniques often neglect 'missingness' information and fail to explicitly model feature and sample correlations, leading to suboptimal results. M3-Impute innovatively incorporates missingness information and correlations through advanced masking schemes. It represents data as a bipartite graph and utilizes a graph neural network with a refined initialization process to learn node embeddings. These embeddings are further optimized using feature correlation and sample correlation units, which explicitly consider the correlations during imputation. The method's effectiveness is demonstrated through experiments on 15 benchmark datasets with three different missing patterns.

**Strengths:**

The feature correlation unit (FCU) and sample correlation unit (SCU) are particularly compelling. The FCU learns correlations between the target missing feature and observed features within each sample, refined by a soft mask on missingness information. Similarly, the SCU computes sample-wise correlations, enhanced by another soft mask on missingness information for pairs of samples.

Integrating FCU and SCU outputs to estimate missing values is methodologically sound. Extensive experiments on 15 open datasets show M3-Impute's superior performance in 13 out of 15 cases under various missing value patterns. The reported improvements in mean absolute error (MAE), up to 11.47% over the second-best method, underscore M3-Impute's practical relevance and robustness.

**Weaknesses:**

M3-Impute demonstrates strong performance across many datasets but shows limitations in handling datasets with highly independent features or strong linear correlations, such as the KIN8NM and NAVAL datasets.

The model's robustness in general scenarios may degrade when confronted with datasets containing extreme correlation structures.

Future improvements could concentrate on enhancing M3-Impute's adaptability to these challenging cases to broaden its applicability and robustness across diverse datasets.

**Questions:**

In ablation study (4.3), Could you provide more details on the refined initialization process of feature-node and sample-node embeddings? How does it differ specifically from the initialization used in Grape?


Can you provide more insights into the computational complexity and runtime performance of M3-Impute compared to the baselines, especially for large datasets?

In missing ratio test, why does M3-Impute perform similarly to Grape on the KIN8NM dataset, and what characteristics of KIN8NM contribute to this result?

**Limitations:**

Although M3-Impute outperforms other baselines on most datasets, two cases (KIN8NM, NAVAL dataset) highlight its limitations in handling datasets with either highly independent features or strongly linear correlations. This suggests that while M3-Impute is robust in general scenarios, its performance may degrade in datasets with extreme correlation structures.

---

> ### Author Rebuttal · Authors · 2024-08-07
>
> We thank the reviewer for recognizing that our FCU and SCU are particularly compelling, our method is methodologically sound, and the experiments are extensive. We also appreciate the reviewer for the constructive comments. Below we provide our response to the concerns raised.
>
>  **Q1. M3-Impute demonstrates strong performance across many datasets but shows limitations in handling datasets with highly independent features or strong linear correlations, such as the KIN8NM and NAVAL datasets. The model's robustness in general scenarios may degrade when confronted with datasets containing extreme correlation structures.**
>
> Thanks for the comment. While our method does not show a significant improvement compared to state-of-the-art methods in such extreme cases, it does remain competitive. For instance, in the KIN8NM and NAVAL datasets, M$^3$-Impute achieves second-best results, with a difference of less than 0.02 in MAE compared to the top performers. We have considered 10 new open datasets in our experiments, in addition to the original 15 datasets, leading to a total of 25 open datasets. Our method consistently outperforms the baselines, demonstrating its general applicability.
>
>  **Q2. Future improvements could concentrate on enhancing M3-Impute's adaptability to these challenging cases to broaden its applicability and robustness across diverse datasets.**
>
> Thanks for the comment. We will explore challenging cases where data have extremely strong or weak correlations in future work.
>
>  **Q3. In ablation study (4.3), Could you provide more details on the refined initialization process of feature-node and sample-node embeddings? How does it differ specifically from the initialization used in Grape?**
>
> We appreciate the opportunity to clarify our approach regarding node embedding initialization.
>
> **Feature-node Embeddings:** In Grape, all feature-node embeddings are initialized as one-hot vectors. Since one-hot vectors are orthogonal, this implicitly assumes that all features are independent. However, in reality, features are often correlated, and initializing feature nodes as one-hot vectors can hinder the modeling of these correlations. To address this issue, we have refined the initialization of feature-node embeddings by using a learnable vector for each feature node. This approach enables the feature node embeddings to be learned during training, allowing correlated features to potentially have similar embeddings from the early stage of learning, which better captures the relationships between features.
>
> **Sample-node Embeddings:** In Grape, sample-node embeddings are initially set to all-one vectors, which results in identical embeddings for all sample nodes at the beginning. This approach does not capture the unique missingness information of each sample. In contrast, we propose using Equation (2) for initializing sample-node embeddings, where observable feature values of a sample are kept and a small value $\epsilon$ is attached to unobserved features. This method ensures that the initial embeddings reflect the missingness information that is specific to each sample.
>
> We hope this explanation clarifies the rationales behind our refinements and their importance in improving imputation accuracy.
>
>  **Q4. Can you provide more insights into the computational complexity and runtime performance of M3-Impute compared to the baselines, especially for large datasets?**
>
> Thanks for the suggestion. The running time of each method is now shown in Table 4 in the rebuttal PDF. The results show that our method is both accurate and time-efficient. Among the datasets, **PR**otein, **SP**am, and **LE**tter are large datasets, where our method takes a little more time to achieve higher accuracy, which we believe is worthwhile.
>
>  **Q5. In missing ratio test, why does M3-Impute perform similarly to Grape on the KIN8NM dataset, and what characteristics of KIN8NM contribute to this result?**
>
> As can be seen from Figure 4 in the appendix of the submitted manuscript, the features in KIN8NM are highly independent. Almost all the imputation methods perform similarly to Grape on this dataset. Please also refer to our response to Q1. Thanks.

---

### Official Review · Reviewer_f6SV · 2024-07-10

**Soundness:** 3
**Presentation:** 3
**Contribution:** 3
**Rating:** 6
**Confidence:** 4

**Summary:**

The paper proposed a new imputations method called M3-impute. M3-impute follows the basic structure of some recent imputation methods: a undirected bipartite graph is constructed with nodes for features and samples, where edge weights correspond to observed data at the given feature-sample pair. Previous approaches use Graph Neural Networks (GNNs) to impute missing values via edge weight prediction. M3-impute improves these approaches by adding two new components on top of an initial GNN to model feature-wise and sample-wise correlations respectively. Empirical results show that M3-impute achieves competitive performance in terms of MAE for imputation across several tabular datasets.

**Strengths:**

- The paper is generally well written.
- Empirical results are extensive. Many other imputations methods are included for comparison, providing a good representation for the state-of-the-art for tabular data imputation. Ablation studies and robustness studies also further strengthen the credibility of the methodology.

**Weaknesses:**

- The paper does not support categorical features. This is a big weakness compared to other imputation methods that can handle categorical features such as iterative approaches like hyperimpute.
- The paper does not discuss the impact of missing value imputation on downstream tasks. Imputation is usually a preprocessing step, and thus assessing the impact on possible downstream tasks is paramount. For example, in supervised learning, some recent evidence suggests that mean/zero imputation is as good as more complex imputations [1, 2].

[1] Le Morvan, Marine, et al. "What’s a good imputation to predict with missing values?." Advances in Neural Information Processing Systems 34 (2021): 11530-11540.
[2] Van Ness, Mike, and Madeleine Udell. "In defense of zero imputation for tabular deep learning." NeurIPS 2023 Second Table Representation Learning Workshop. 2023.

**Questions:**

- What is the current approach for handling categorical features? Do none of the datasets in the experiments have any categorical features, or are these feature simply being one-hot encoded?
- How does the runtime of M3-impute compare to the other methods? I know this is briefly discussed in appendix 5, but more details runtimes would be appreciated.
- Why are standard deviations not included for Table 1?

**Limitations:**

See weaknesses. In particular, not handling categorical features is not mentioned in the paper anywhere as a limitation of the method.

---

> ### Author Rebuttal · Authors · 2024-08-07
>
> We sincerely thank you for recognizing that our paper is well-written and  our empirical results are extensive. Below we provide our response to the concerns raised.
>
>  **Q1. The paper does not support categorical features. This is a big weakness compared to other imputation methods that can handle categorical features such as iterative approaches like hyperimpute.**
>
> We appreciate your feedback and would like to clarify that our method is indeed capable of handling mixed features, including categorical ones.
>
> Specifically, during the forward computation, in M$^3$-Impute, we first convert categorical feature values into numerical values (e.g., categories 1, 2, and 3 are converted to real numbers 1, 2, and 3) and incorporate them alongside numerical features in the initialization stage. Subsequently, M$^3$-Impute performs GNN computations to obtain node embeddings. These embeddings are then processed through Feature Correlation Unit (FCU) and Sample Correlation Unit (SCU) to compute the corresponding context vectors. Finally, M$^3$-Impute uses an MLP with a ReLU activation function for imputing numerical features and an MLP with a softmax activation function for imputing categorical features. This is feasible because we know the data type of the target feature to be imputed and can switch the activation function accordingly. For parameter updates, we use the L1 error for numerical features and the cross-entropy loss for categorical features.
>
> In our submitted manuscript, we already reported the results on the performance of our method on the datasets with mixed feature types, such as the housing and airfoil datasets. To further demonstrate the efficacy of our method in handling categorical features, we have evaluated M$^3$-Impute on four new datasets with mixed feature types. The dataset statistics are presented in Tab.1 and the results are shown in Tab.2.
>
> ### Tab.1: 4 Additional datasets that contain categorical features.
>
> |                         | abalone | Ai4i  | CMC  | German |
> |-------------------------|:---------:|:-------:|:------:|:--------:|
> | #Samples             | 4177    | 10000 | 1473 | 1000   |
> | #Numerical Features  | 7       | 7     | 8    | 13     |
> | #Categorical Features| 1       | 5     | 1    | 7      |
> |
>
> ### Tab.2: Imputation accuracy in MAE under MCAR setting with 30\% missingness.
>
> |            | Abalone | Ai4i | CMC  | German |
> |------------|:---------:|:------:|:------:|:--------:|
> | Mean   | 2.52    | 1.07 | 2.35 | 2.52   |
> | Svd    | 2.60    | 1.18 | 2.52 | 2.60   |
> | Spectral | 2.45  | 1.58 | 2.96 | 2.45   |
> | Mice   | 2.26    | 0.87 | 2.06 | 2.26   |
> | kNN    | 2.34    | 1.17 | 2.32 | 2.34   |
> | Gain   | 2.27    | 1.03 | 2.33 | 2.27   |
> | Miwae  | 2.59    | 1.12 | 2.37 | 2.59   |
> | Grape  | *2.01*  | 0.79 | *1.87* | *2.01* |
> | Miracle| 39.17   | 1.02 | 2.16 | 39.17  |
> | HyperImpute | 2.05 | **0.75** | 1.91 | 2.05   |
> | **M³-Impute** | **1.84** | *0.76* | **1.81** | **1.87** |
> |
>
>  **Q2. The paper does not discuss the impact of missing value imputation on downstream tasks. Imputation is usually a preprocessing step, and thus assessing the impact on possible downstream tasks is paramount. For example, in supervised learning, some recent evidence suggests that mean/zero imputation is as good as more complex imputations.**
>
> Thanks for your constructive comments. We have conducted new experiments on the impact of imputation methods on downstream tasks and report the results in Tab.3. The results show that M$^3$-impute outperforms the baselines, including the mean imputation, for most cases. We will include the new results in the final manuscript.
>
> ### Tab.3: Averaged RMSE of label prediction under MCAR setting with 30\% missingness.
>
> |              | Yacht | Wine | Concrete | Housing | Energy | Naval  | Kin8nm | Power |
> |--------------|:-------:|:------:|:----------:|:---------:|:--------:|:--------:|:--------:|:-------:|
> | Mean     | 13.10 | 0.69 | 13.10    | 6.19    | 5.49   | 0.0075 | **0.22** | 8.38  |
> | Svd      | 13.60 | 0.69 | 13.80    | 6.08    | 4.82   | 0.0070 | 0.24   | 9.87  |
> | Spectral | 13.70 | 0.69 | 13.40    | 6.13    | 4.81   | 0.0067 | _0.23_  | 9.45  |
> | Mice     | 12.80 | 0.69 | 13.00    | 5.80    | 3.99   | 0.0057 | **0.22** | 6.94  |
> | kNN      | 13.00 | _0.67_ | 12.40  | 5.85    | 4.02   | 0.0062 | _0.23_  | 7.75  |
> | Gain     | 13.70 | 0.68 | 13.30    | 5.91    | 3.67   | 0.0070 | _0.23_  | 6.83  |
> | Miwae    | 17.40 | 0.70 | 13.20    | 7.28    | 4.98   | 0.0075 | _0.23_  | 8.45 |
> | NeuMiss  | 16.79 | 0.87 | 14.44    | 18.13   | 14.09  | 1.95   | 0.22   | 56.74 |
> | Grape    | _12.67_ | _0.67_ | _11.63_ | _5.23_  | 3.60   | _0.0049_ | _0.23_  | 6.72  |
> | Miracle  | 17.80 | 0.70 | 13.20    | 9.24    | 9.54   | 0.0051 | **0.22** | 6.62  |
> | HyperImpute | 13.10 | _0.67_ | 12.40  | 5.45    | _3.54_ | **0.0043** | _0.23_  | **6.44** |
> | **M${^3}$-Impute** | **12.43** | **0.66** | **11.47** | **5.19** | **3.45** | _0.0049_ | **0.22** | _6.49_  |
> |
>
>  **Q3. How does the runtime of M3-impute compare to the other methods? I know this is briefly discussed in appendix 5, but more details runtimes would be appreciated.**
>
> Thanks for the question. We now report the running time performance of each imputation method in Table 4 in the rebuttal PDF. We will incorporate it into the final manuscript.
>
>  **Q4. Why are standard deviations not included in Table 1?**
>
> Thanks for the question. Due to space constraints, we were unable to include both MAE scores and the standard deviations of all methods in Table 1, but we provided the comprehensive results (including both MAE scores and standard deviations) in Table 8 in the appendix of the submitted manuscript.

---

> > ### Comment · Reviewer_f6SV · 2024-08-08
> >
> > I've raised my score to a 6 (weak accept) as the authors have addressed several of the concerns in my review.

---

### Official Review · Reviewer_LMmS · 2024-07-12

**Soundness:** 3
**Presentation:** 3
**Contribution:** 2
**Rating:** 5
**Confidence:** 4

**Summary:**

This paper proposes M^3-Impute, a missing value imputation method that utilizes GNNs to learn embeddings of samples and features. By incorporating feature correlation unit and sample correlation unit, M^3-Impute effectively captures correlations between features and samples for accurate imputation.

**Strengths:**

S1. This paper introduces a novel masking scheme that effectively utilizes missing information for modeling.

S2. This paper proposes the feature correlation unit (FCU) and sample correlation unit (SCU), which help to consider feature and sample correlations during imputation.

S3. Experimental evaluations on various datasets compare the proposed method with state-of-the-art approaches, demonstrating good imputation performance.

**Weaknesses:**

W1. SCU takes into account the pairwise similarity of \mathcal{P} during its construction, which subsequently determines the scalar parameter \alpha during imputation. The initialization of \mathcal{P} seems to directly impact the model's performance and remains unchanged once set. It would be beneficial for the authors to discuss this aspect and, if possible, provide some experimental evidence to support their approach.

W2. In Section 4.4, the paper mentions different sampling strategies for SCU and uses a new strategy in the ablation study (Table 2), which is different from the strategy mentioned in Section 3.4. The authors claim that this strategy leads to inferior performance compared to previous strategies, thereby highlighting the superiority of the ablation study results. This lacks experimental evidence and results in inconsistency in the experimental setup.

W3. As a crucial parameter, the size of \mathcal{P} directly affects the construction of SCU. Table 3 presents experimental results with different sizes, but the differences are not significant, which is somewhat counterintuitive. Although the authors discuss the experimental results in Section 4.4, the performance fluctuation of 0.01 to 0.02 does not clearly reflect the "decrease then increase" trend mentioned by the authors. Exploring larger peer values and providing more detailed analysis and guidance on parameter selection might be beneficial.

W4. The authors emphasize the importance of specific missingness information throughout the paper. Intuitively, different types of missing data (MAR, MNAR, MCAR) might offer varying types of missingness information, potentially impacting the model's performance. While the paper experiments with data of different missingness types, a more thorough discussion of the results could enhance the motivation and clarity of the paper.

W5. In Section 4.3, the caption of Table 2 references a concept, the uniform sampling strategy, which is introduced for the first time in a later subsection. The authors might consider adjusting the structure of the paper for better clarity.

W6. In Figure 3, in the subfigure with the missing ratio of 0.7, two bars exceed the upper boundary and need adjustment.

W7. It would be helpful to add independent labels to all subfigures, such as (a), (b), etc., to facilitate referencing.

**Questions:**

Q1. In Section 3.4, the authors mention that \mathcal{P} is constructed by randomly choosing with a certain probability. Why not select directly based on the degree of similarity rather than introducing randomness? How might these different approaches impact the model's performance?

Q2. To what extent does applying different GNN models to learn embeddings impact the performance of the model proposed in this paper?

**Limitations:**

Yes.

---

> ### Author Rebuttal · Authors · 2024-08-07
>
> We sincerely thank the reviewer for recognizing that our masking schemes are novel, the proposed correlation units are helpful, and our experiments demonstrate good performance. We also appreciate the constructive comments. Below we provide our response to the concerns raised.
>
>  **Q1. The initialization of $\mathcal{P}$ in SCU remains unchanged once set. Why not select peers directly based on the degree of similarity rather than introducing randomness?**
>
> Sorry for the confusion caused. $\mathcal{P}$ is obtained from a sampling strategy where the probability of a peer being selected is proportional to its cosine similarity with the target node. The set $\mathcal{P}$ is updated every epoch and does *not* remain unchanged.
>
> We would also like to point out that the set $\mathcal{P}$ balances both high-similarity peers and potential peers that serve for regularization and generalization purposes since directly selecting peers based on the degree of similarity, such as using a $k$-nearest neighbors method, may introduce extra computational complexity and hinder the method's generalization ability. We will incorporate the clarification into our final manuscript. Thanks.
>
>  **Q2. Table 2 in Section 4.4 for sampling strategy comparison is confusing; the caption is unclear.**
>
> The cosine-similarity sampling strategy (introduced in Section 3.4) is integrated into M$^3$-impute for the main results across all experiments. Section 4.4 presents an ablation study where the original cosine-similarity sampling strategy is replaced with a uniform sampling strategy. The performance comparison is shown in Table 2, where M$^3$-impute indicates the cosine-similarity sampling strategy and M$^3$-uniform represents the uniform sampling strategy. As uniform sampling does not consider peer similarities, its performance is expected to be inferior to the original strategy, which is verified in Table 2. We will elaborated more in the table caption and incorporate the changes into the final manuscript.
>
>  **Q3. Table 3 does not clearly present the trend of $\mathcal{P}$.**
>
> As explained in our response to Q1, the set $\mathcal{P}$ is updated every epoch. A proper peer size should balance high-similarity peers and potential peers that serve for regularization and generalization purposes. In general, the trend across different datasets shows that a too-small peer size may only include high-similarity peers, while a too-large peer size may include too many noisy nodes and incur higher computational overhead. The small fluctuations indicate that our method is relatively robust to this parameter. From the extensive experiments on 25 datasets, we recommend a peer size of 5–10 for practical use. We will properly revise the manuscript, not only proving a more comprehensive table but also showing the running time. Thanks.
>
>  **Q4. A more thorough discussion of the results under different missing types could enhance the motivation and clarity of the paper.**
>
> Thanks for the comments. As demonstrated in Tables 1, 5, and 6 in our manuscript, M$^3$-Impute consistently outperforms the baselines across all three missingness patterns. This superior performance is due to M$^3$-Impute's unique approach. Rather than assuming the data follows MCAR, MAR, or MNAR missingness patterns from the outset, we designed M$^3$-Impute to leverage missingness information directly, enabling it to learn feature-wise and sample-wise correlations. Specifically, in Feature Correlation Unit (FCU), M$^3$-Impute uses the missingness results (i.e., known masks) to learn the correlations between the imputation targets and observable features. In Sample Correlation Unit (SCU), M$^3$ employs the missingness results to better capture sample correlations. Since feature and sample correlations exist regardless of the cause of missingness, M$^3$-Impute is naturally adaptive and robust across all three missingness settings.
>
> Another notable feature of M$^3$-Impute is its ability to learn the cause of missingness. In the FCU unit, M$^3$-Impute explicitly captures the correlations between observed and missing features. When data is missing under MAR and MNAR conditions, the missing values depend on the observed ones. Since FCU explicitly captures these relationships, M$^3$-Impute can potentially identify the cause of missingness and enhance imputation accuracy. This capability may explain why M$^3$-Impute significantly outperforms the baselines in the MAR and MNAR settings compared to the MCAR settings.
>
>  **Q5. Two bars exceed the upper boundary of Figure 3.**
>
> We will update the figure to show the full range of the performance. Thanks.
>
>  **Q6. Add independent labels to all subfigures.**
>
> The subfigure labels will be added to the final manuscript. Thanks.
>
>  **Q7. The influence on performance by applying different GNN models.**
>
> We have conducted new experiments using GNN variants such as GraphSAGE, GAT, and GCN. The results are shown in Table 3 in the rebuttal PDF. The results indicate that different aggregation mechanisms may introduce varying errors, but our method consistently outperforms its GRAPE counterpart, demonstrating its effectiveness.

---

> > ### Comment · Reviewer_LMmS · 2024-08-10
> >
> > Thank you for the response. My concerns have been addressed, and I will raise my score to 5.

---

### Official Review · Reviewer_JuhZ · 2024-07-12

**Soundness:** 3
**Presentation:** 3
**Contribution:** 3
**Rating:** 7
**Confidence:** 3

**Summary:**

This is a novel approach for imputing missing data using mask-guided representation learning. The main contributions include the development of an imputation model that leverages both feature and sample correlations. This model improves imputation accuracy and robustness compared to existing methods. The paper also provides comprehensive experiments and ablation studies to validate the effectiveness of the proposed approach across various datasets and missing data scenarios.

**Strengths:**

- Novelty: A unique mask-guided representation learning method that effectively combines feature-wise and sample-wise correlations.
- Comprehensive experiments
- Strong empirical performance

**Weaknesses:**

- Computational complexity

## Minor Points

l4: "Existing imputation methods, however, fall short of considering the ‘missingness’ information in the data  during initialization and modeling the entangled feature and sample correlations  explicitly during the learning process,"
-> This is not true. Many existing methods consider missingness patterns.

The distinction between "statistical" and "learning based" methods seems off. Certainly most learning based methods are statistical, and vice versa.

l. 38 "struggles" -> struggle

**Questions:**

- (How) do you do HPO for competing methods?

**Limitations:**

- Gives little insight into why it works better on some datasets than others.
- Would be interesting to understand better how robust results are under systematic changes in datasets, e.g., different types of missingness.

---

> ### Author Rebuttal · Authors · 2024-08-07
>
> We sincerely thank the reviewer for recognizing that our work is novel with a unique mask-guided representation learning method and we demonstrate strong empirical performance with comprehensive experiments. Below we provide our response to the concerns raised.
>
>  **Q1. Weakness: Computational complexity**
>
> Thanks for the comment. The nature of neural network architectures makes it difficult to analyze the computational complexity rigorously. Nonetheless, we provide numerical results of running time here to show the computation overhead of each method. Please refer to Table 4 in the rebuttal PDF.
>
>  **Q2. Minor Points \#1: ''l4: `Existing imputation methods, however, fall short of considering the ‘missingness’ information in the data during initialization and modeling the entangled feature and sample correlations explicitly during the learning process,' $\rightarrow$ This is not true. Many existing methods consider missingness patterns.''**
>
> We appreciate your feedback. Regarding the point about ''fall short of considering the missingness information in the data during initialization'' not being precise, our intention was to highlight that many existing methods ignore the missingness information during their **initialization stage**. For instance, in iterative imputation frameworks such as MICE and HyperImpute, the initialization stage often involves filling in missing values with some starting values (typically using the mean values of features) before learning. Similarly, graph-based imputation methods like GRAPE and IGRM do not explicitly consider missingness information when initializing their node and edge embeddings. Nonetheless, we agree that our statement can be misleading. We will carefully revise the statement in the final manuscript to avoid any confusions. Thanks.
>
>  **Q3. Minor Points \#2: ''The distinction between statistical and learning based methods seems off. Certainly most learning based methods are statistical, and vice versa.''**
>
> Thanks for the suggestion. We agree with this point and will restructure our related work section in the final manuscript.
>
>  **Q4. Minor Points \#3: ''l. 38 struggles $\rightarrow$ struggle''**
>
> Thank you very much for pointing out the typo. We will correct it in the final version.
>
>  **Q5. Questions: (How) do you do HPO for competing methods?**
>
> For baselines, we followed the commonly used hyperparameter settings from the previous studies, such as GRAPE, including edge drop-out ratio during training, dropout rate, learning rate, and the number of GNN layers. For M$^3$-Impute, we used the same set of parameters for all the experiments. Additionally, we have now included an analysis of the impact of hyperparameters on the performance of M$^3$-Impute. Results can be found in Tables 5-7 in the rebuttal PDF. They will be incorporated into the final manuscript. Thanks.
>
>  **Q6. Limitation1: ''Gives little insight into why it works better on some datasets than others.''**
>
> Thanks for the comment. We included a performance analysis on different datasets in `Section 4.2 Overall Performance' of the submitted manuscript. To summarize, we apply learnable masks to the data matrix with missing values via our novel units, Feature Correlation Unit (FCU) and Sample Correlation Unit (SCU) to better capture feature-wise and sample-wise correlations, respectively. These correlations are crucial for improving the accuracy of missing data imputation.
>
> We admit that there are few datasets with extreme cases of correlations, e.g., having almost all independent features or all completely dependent features, where our method may not perform as good as on the other datasets. For instance, in the KIN8NM dataset where most features are independent of each other, M$^3$-Impute does not perform as effective as it can be. It is somewhat expected since knowing any of the non-missing features offers little help in imputing the missing ones due to their independence. Nonetheless, M$^3$-Impute achieves second-best results for such datasets, and it does remain competitive, with a difference of less than 0.02 in MAE compared to the top performers. Furthermore, we have considered 10 new open datasets in the experiments, together with the original 15 datasets, leading to a total of 25 open datasets. The results show that our method consistently outperforms the baselines for most cases, demonstrating its general applicability.
>
>  **Q7. Limitation2: ''Would be interesting to understand better how robust results are under systematic changes in datasets, e.g., different types of missingness.''**
>
> Thanks for the comment. We indeed reported the performance of our method under three types of missingness in the submitted manuscript. The results for the MAR and MNAR settings are presented in Table 5 and Table 6 in the appendix of the submitted manuscript, respectively. In addition to the 8 UCI datasets discussed in the main text, we also included performance results on 7 additional datasets under all three types of missingness. These results can be found in Table 9 of the appendix of the submitted manuscript.

---

> > ### Comment · Reviewer_JuhZ · 2024-08-12
> > **Thanks for the detailed response! My rating remains.**
> >
> > .

---

### Official Review · Reviewer_LUcW · 2024-07-12

**Soundness:** 3
**Presentation:** 3
**Contribution:** 2
**Rating:** 6
**Confidence:** 3

**Summary:**

This paper presents a novel imputation method, based on a bipartite graph constructed from the data and the missing-data patterns, and two components which allow to measure similarities between the features and samples.

The method shows very good results in terms of MAE on several datasets for MCAR, MAR and MNAR data.

**Strengths:**

- The paper is well written.

- Experiments are well conducted, on several datasets, with different missing-data ratios and considering MCAR, MAR or MNAR data. The authors have made an effort to compare themselves with many other imputation methods.

- There is a true discussion on the parameters to choose in the experiments. The authors are honest about the performance of their method, and give explanations when another method is better.

**Weaknesses:**

- Although well presented, the method is complicated to understand.

- The methods uses 8 MLPs and one GNN. The authors discuss in Appendix the computational resources, but do not compare other methods on this point.

**Questions:**

General remarks:
- How does this methodology relate to the simple concatenation of the mask to the data matrix, and the execution of an imputation method on the augmented matrix? (see Josse, Julie, et al. "On the consistency of supervised learning with missing values.")
- Is M3-Impute supposed to work well for MNAR? This should be discussed more in details, as the authors claim that the method utilizes the data-missingness information. A remark: there exists for MIWAE an extension specifically designed for MNAR data, called not-MIWAE. Ipsen, Niels Bruun, Pierre-Alexandre Mattei, and Jes Frellsen. "not-MIWAE: Deep generative modelling with missing not at random data."
It can be interesting to have a comparison of M3-Impute with this one in a final version.

Algorithm:
- Figure 1: maybe the authors should add numbers in the graphics to refer to them when describing the method in the text (especially in 3.1)
- In Algorithm 1, one of the input is the GNN model. How are hyperparameters of the GNN managed in practice?

Numerical experiments:
- In the final version, the authors should add a comparison with the missForest algorithm, which is one of the most widely used imputation methods.
- for other methods, such as MIWAE, which hyperparameters did the authors choose?
- in the MAR setting, how many features are selected to be observed? Did the authors take the best subset for the results?

Minor comments:
- l.15 mechanisms instead of patterns
- l.56 "the the"
- l.137 notation col_s: harmonise d and m
- l.548 "these remaining" <- "the remaining"
- l.291 mechanisms instead of patterns

**Limitations:**

Yes

---

> ### Author Rebuttal · Authors · 2024-08-07
>
> We thank the reviewer for recognizing our paper as well-written and the experiments as well-conducted. We also appreciate the constructive comments. Below we provide our response to the concerns raised.
>
> **Q1. Although well presented, the method is complicated to understand.**
>
> The main idea behind our method is to explicitly utilize the missingness information as input and apply learnable masks to the data matrix to better capture feature-wise and sample-wise correlations, thereby improving  imputation accuracy. To this end, we propose two novel units, Feature Correlation Unit (FCU) and Sample Correlation Unit (SCU), to capture feature-wise and sample-wise correlations, respectively. We will improve the presentation of the final manuscript. Thanks.
>
> **Q2. Computation resource comparison.**
>
> We have added the comparison of running time with other methods in Tab.4 of the rebuttal PDF and will incorporate this into the final manuscript.
>
> **Q3. Difference from simple concatenation of the mask to the data matrix.**
>
> We would like to point out that the simple concatenation of the mask to the data matrix corresponds to our transformation of the masked data matrix into a bipartite graph, and the execution of an imputation method on the augmented matrix corresponds to running our M$^3$-Impute method on the bipartite graph.
>
> In addition, while we are not quite sure in what sense the paper ''On the consistency...'' is referred to here, we point out that M$^3$-Impute is a task-general architecture, as it is not limited to any specific downstream task. In contrast, in their paper, a downstream task is involved in the design of their method. We further compare M$^3$-Impute with theirs (NeuMiss + MLP). As shown in Tab.1, M$^3$-Impute consistently outperforms. We will include the new results in the final manuscript. Thanks.
>
> ### Tab.1: MAE of label prediction under MCAR setting with 30\% missingness
>
> |          | Yacht | Wine | Concrete | Housing | Energy | Naval | Kin8nm | Power |
> |----------|:-------:|:------:|:----------:|:---------:|:--------:|:-------:|:--------:|:-------:|
> | NeuMiss+MLP  | 11.69 | 0.65 | 11.57    | 14.72   | 11.04  | 1.25  | **0.18**   | 27.57 |
> | M${^3}$-Impute | **8.82** | **0.51** | **9.04** | **3.60** | **2.57** | **0.0036** | **0.18** | **4.69** |
> |
>
> **Q4. Performance under MNAR setting and comparison with not-MIWAE.**
>
> We presented the results under the MNAR setting in Table 6 in the submitted manuscript and confirmed that M$^3$-Impute consistently outperforms the baselines. This superior performance is due to M$^3$-Impute's unique approach. Rather than assuming the data follows MCAR, MAR, or MNAR missingness patterns from the outset, we designed M$^3$-Impute to leverage missingness information directly, enabling it to learn feature-wise and sample-wise correlations. Specifically, in FCU, M$^3$-Impute uses the missingness results (i.e., known masks) to learn the correlations between the imputation targets and observable features. In SCU, M$^3$ employs the missingness results to better capture sample correlations. Since feature and sample correlations exist regardless of the cause of missingness, M$^3$-Impute is naturally adaptive and robust across all three missingness settings.
>
> Another notable feature of M$^3$-Impute is its ability to learn the cause of missingness. In FCU, M$^3$-Impute explicitly captures the correlations between observed and missing features. When data is missing under MAR and MNAR conditions, the missing values depend on the observed ones. Since FCU explicitly captures these relationships, M$^3$-Impute can potentially identify the cause of missingness and enhance imputation accuracy. This capability may explain why M$^3$-Impute significantly outperforms the baselines in the MAR and MNAR settings compared to the MCAR settings.
>
> In addition, we have done new experiments for the comparison with not-MIWAE and report the results in Tab.2, showing that our method outperforms not-MIWAE substantially. We will include the results in the final manuscript. Thanks.
>
> ### Tab 2: MAE of imputation under MNAR setting with 30\% missingness
>
> |            | Yacht | Wine | Concrete | Housing | Energy | Naval | Kin8nm | Power |
> |------------|:-------:|:------:|:----------:|:---------:|:--------:|:-------:|:--------:|:-------:|
> | not-MIWAE  | 3.08  | 1.43 | 2.14     | 1.80    | 3.87   | 2.27  | 2.50   | 2.46  |
> | M³-Impute  | **1.15** | **0.60** | **0.68** | **0.54** | **1.09** | **0.08** | **2.46** | **1.00** |
> |
>
> **Q5. Add numbers to Figure 1.**
>
> The numbers will be included in the final manuscript. Thanks.
>
> **Q6. Hyperparameter setting in Algorithm 1 and other baselines.**
>
> For all the 25 datasets, we use the same hyperparameters for our method and follow the same setups as in the original papers for the baselines.
>
> **Q7. Comparison with missForest.**
>
> The results are shown in Tab.3 and will be added in the final manuscript. Thanks.
>
> ### Tab. 3: MAE of imputation under MCAR setting with 30\% missingness
>
> |           | Yacht | Wine | Concrete | Housing | Energy | Naval | Kin8nm | Power |
> |-----------|:-------:|:------:|:----------:|:---------:|:--------:|:-------:|:--------:|:-------:|
> | MissForest| 1.78  | 0.73 | 1.31     | 0.80    | 1.48   | 0.25  | 2.52   | 1.18  |
> | M³-Impute | **1.33** | **0.60** | **0.71**  | **0.59** | **1.31** | **0.06** | **2.50**| **0.99** |
> |
>
> **Q8. Feature selection in MAR setting.**
>
> For the 30\% missingness setup, we randomly selected 50\% of the features to be observed (ensuring these features do not contain any missing values) and masked out values from the remaining 50\% of the features until the desired missingness ratio is reached. We did not cherry-pick the subset of observed features; rather, we randomly selected them so that they could be different in each repeated run, as different random seeds are applied.
>
> **Q9. Minor comments:**
>
> They will be incorporated in the final manuscript. Thanks.

---

### Author Rebuttal · Authors · 2024-08-07

We appreciate the constructive comments from Reviewer LUcW (R1), JuhZ (R2), LMmS (R3), f6SV (R4), E1Qa (R5), Bqr5 (R6), and WjjR (R7). We are encouraged that they find our approach novel (R1, R2, R3, R5, R6, R7), our masking scheme innovative and effective (R2, R3, R5, R7), our experiments comprehensive and extensive (R1, R2, R4, R5, R7), and our manuscript well written (R1, R4). Below, we address the common concerns raised. For individual questions, please refer to our separate response to each reviewer.

**Experiments on 10 more datasets:**

We have further tested our method on 10 additional datasets, totaling 25 datasets. Of the 10 new datasets, five are relatively large, and four contain mixed types of features. Details of the datasets and the results are provided in Table 1 and Table 2 below, respectively. The results again demonstrate the effectiveness of M$^3$-Impute, achieving nine best and one second-best in imputation accuracy.

**Table 1: 10 additional datasets for data imputation: **PR**otein, **SP**am, **LE**tter, **AB**alone, **AI**4i, **CM**c, **GE**rman, **ST**eel, **LI**bras, and **CA**lifornia-housing, totaling 25 datasets studied.**
|            | PR    | SP   | LE    | AB   | AI   | CM   | GE   | ST   | LI  | CA    |
|------------|:-------:|:------:|:-------:|:------:|:------:|:------:|:------:|:------:|:-----:|:-------:|
| # of Samples    | 45730 | 4601 | 20000 | 4177 | 10000| 1473 | 1000 | 1941 | 360 | 20640 |
| # of Features| 9    | 57   | 16    | 8    | 12   | 9    | 20   | 33   | 91  | 9     |
|

**Table 2: MAE under MCAR setting with 30\% missingness. Please refer to the caption of Table 1 for dataset names.**
| Model         | PR    | SP    | LE    | AB    | AI    | CM    | GE    | ST    | LI    | CA    |
|---------------|:-------:|:------:|:-------:|:------:|:------:|:------:|:------:|:------:|:-----:|:-------:|
| Mean          | 0.91  | 0.23  | 1.28  | 2.52  | 1.07  | 2.35  | 2.52  | 1.80  | 1.82  | 1.13  |
| Svd           | 1.00  | 0.31  | 1.29  | 2.60  | 1.18  | 2.52  | 2.60  | 1.37  | 0.37  | 1.35  |
| Spectral      | 1.14  | **0.16** | 1.75  | 2.45  | 1.58  | 2.96  | 2.45  | 1.10  | 0.18  | 1.50  |
| Mice          | 0.33  | 0.22  | 1.00  | 2.26  | 0.87  | 2.06  | 2.26  | 0.95  | _0.11_ | 0.69  |
| kNN           | 0.58  | _0.17_ | 0.89  | 2.34  | 1.17  | 2.32  | 2.34  | 0.78  | 0.25  | 1.17  |
| Gain          | 0.72  | 0.21  | 1.09  | 2.27  | 1.03  | 2.33  | 2.27  | 1.03  | 0.46  | 1.07  |
| Miwae         | 0.94  | **0.16** | 1.33  | 2.59  | 1.12  | 2.37  | 2.59  | 1.70  | 2.13  | 1.16  |
| Grape         | _0.25_ | _0.17_ | _0.53_ | _2.01_ | 0.79  | _1.87_ | _2.01_ | _0.45_ | **0.10** | **0.54** |
| Miracle       | 0.32  | 1.07  | 1.06  | 39.17 | 1.02  | 2.16  | 39.17 | 1.51  | 51.37 | 0.67  |
| HyperImpute   | _0.25_ | 0.18  | 0.61  | 2.05  | **0.75** | 1.91  | 2.05  | 0.72  | _0.11_ | _0.57_ |
| **M${^3}$-Impute** | **0.24** | **0.16** | **0.52** | **1.84** | _0.76_ | **1.81** | **1.87** | **0.39** | **0.10** | **0.54** |
|

**Computational resources and runtime (@R1, R2, R5, R6):**

We have added a runtime comparison in Table 4 in the rebuttal PDF and will include the results in the final manuscript. The results show that our method is both accurate and time-efficient. For example, for inference with GPU, the time taken to impute *all* the missing values for any dataset we tested is less than one second under the setting of MCAR with 30\% missingness.

**Performance on downstream tasks (@R1, R4):**

We have conducted new experiments on the impact of imputation methods on downstream tasks. Specifically, given datasets with missing values, we first impute all the missing values and then use the completed datasets for downstream tasks. We report the results on the downstream-task performance of imputation methods in Tab.1 in our response to @R1 and Tab.3 in our response to @R4. As shown in the tables, our method consistently outperforms the imputation baselines, indicating that the values imputed by our method are more beneficial for downstream tasks.

**Hyperparameter management (@R1, R7):**

We followed the commonly used hyperparameter settings from the previous studies for baselines. For M$^3$-Impute, we used the same set of parameters for all the experiments. Additionally, we have now included an analysis of the impact of hyperparameters on the performance of M$^3$-Impute. Results can be found in Tables 5-7 in the rebuttal PDF.

**Questions regarding the set $\mathcal{P}$ (@R3)**

The set $\mathcal{P}$ is obtained through a sampling strategy based on the cosine similarity with the target node since directly selecting peers based on the degree of similarity, e.g., using $k$-nearest neighbors, may introduce extra computational complexity and hinder the method's generalization ability. It is *updated* every epoch. We also provide a rule of thumb for the size of the set based on extensive experiments. Please refer to our responses to Q1 and Q3 @R3 for more details.

**Categorical Features Handling (@R4):**

We would like to point out that our method M$^3$-Impute can handle mixed types of features, including categorical ones. Specifically, it converts categorical features into numerical values during initialization and uses an MLP with softmax activation for imputing categorical values, with cross-entropy loss for parameter updates. We already included the results of M$^3$-Impute on the datasets with mixed feature types, which are shown in Tables 1 and 9 in our submitted manuscript. We have further tested M$^3$-Impute on four additional datasets with mixed data types. Dataset details and results can be found in Tab.1 and Tab.2 in our response to @R4.

**Imputation results with RMSE (@R7):**

We have included new results on the performance of imputation methods measured in RMSE in Tab.1 in our response to @R7. The results show that our method consistently outperforms the baselines. We will include them in the final manuscript.

---

### Author Response · Authors · 2024-08-14
**Thanks for the comments.**

We would like to thank all the reviewers again for their invaluable time and constructive comments. We will incorporate all the corresponding updates into our final manuscript. Thank you.

---

### Decision · Program_Chairs · 2024-09-25

**Decision:**

Reject

**Comment:**

The authors propose a new imputation algorithm based on deep learning. The starting point is to see the incomplete data matrix as a bipartite graph, with "feature" nodes and "samples" nodes, than can then be handled by a graph neural network to produce imputations.

All reviewers agreed that the method is clever and interesting, in particular in its ability to process the missingness pattern. Some argued that it was a bit overly complex, and I tend to agree (in particular, I would suggest the authors to write a more detailed caption for Fig.1, that would explain the pipeline concisely).

Most of the reviewers's concerns were answered. In my opinion, there is a major point that remains. Two reviewers asked for more details about hyper-parameter optimisation of competing methods, and the authors's response was vague and unconvincing:

> For baselines, we followed the commonly used hyperparameter settings from the previous studies, such as GRAPE, including edge drop-out ratio during training, dropout rate, learning rate, and the number of GNN layers.

It seems clear that hyperparameter optimisation was poorly done for some of the methods. Indeed, it seems virtually impossible that methods that contain the mean imputation as a special case (like SVD or MIWAE) can do worse than the mean imputation when hyperparameters are properly optimised (SVD is often worse than the mean in the experiments, and MIWAE is almost always worse). Moreover, hyperparameter optimisation was also very problematic for MIRACLE: e.g. in Table 1, for 3 datasets, the MAE of MIRACLE is more than *ten times* the MAE of the mean imputation. For future iterations, I strongly encourage the authors to revisit their choices of hyperparameters to produce fairer comparisons. In particular, for methods based on deep nets (like MIWAE or MIRACLE), I would suggest carefully looking at how the data were normalised. Methods that produce results worse than mean imputation should be investigated and discussed, since this means exceptionally bad performance.

Another point that should be mentioned is that most of these methods were designed to minimise the MSE, not the MAE. The authors also conducted experiments on the MSE during the rebuttal, but experiments with the MAE should clearly indicate which method was designed to minimise the MSE, and which method was designed to minimise the MSE.

After an extensive discussion with one of the reviewers who recommended acceptance and the senior AC, we agreed to reject the paper on these grounds. In future iterations, I encourage the authors to focus on explaining the method in a clearer fashion, and making fairer comparisons, with clear choices of hyper-parameters and network architectures.

---

> ### Public Comment · ~Weipeng_Zhuo1 · 2024-11-29
>
> Again, we would like to thank all the reviewers and the AC for their invaluable time and constructive comments. We would like to provide our response below to the concerns that were newly raised after the discussion period.
>
> (1) For the hyperparameters of the baselines, we used the optimal settings from Grape [1] and HyperImpute codebase [2] for a fair comparison. Though we agree that "Methods that produce results worse than mean imputation should be investigated and discussed, since this means exceptionally bad performance", we observe a similar trend of some baseline methods performing worse than the mean imputation in other works [1,2], which may be due to the small number of features or samples.
>
> As suggested by the AC, we conducted a hyperparameter search for the baseline models, especially for MIWAE and MIRACLE. Results show that MIRACLE's performance improves substantially with a small embedding dimension, while MIWAE's performance improves with more iterations of training, albeit marginally. Despite such a hyperparameter tuning, the proposed method M
> -impute still achieves lowest MAE scores on average compared to all the other imputation methods. We have included the updated experiment results below, which are obtained after the hyperparameter tuning.
>
> [1] You, Jiaxuan, Xiaobai Ma, Yi Ding, Mykel J. Kochenderfer, and Jure Leskovec. Handling missing data with graph representation learning. Advances in Neural Information Processing Systems 33 (2020): 19075-19087.
>
> [2] Jarrett, Daniel, Bogdan C. Cebere, Tennison Liu, Alicia Curth, and Mihaela van der Schaar. Hyperimpute: Generalized iterative imputation with automatic model selection. In International Conference on Machine Learning, pp. 9916-9937. PMLR, 2022.
>
>
> |               | Yacht            | Wine            | Concrete        | Housing         | Energy          | Naval           | Kin8nm          | Power           |
> |---------------|------------------|-----------------|-----------------|-----------------|-----------------|-----------------|-----------------|-----------------|
> | Mean          | 2.09 ± .04       | 0.98 ± .01      | 1.79 ± .01      | 1.85 ± .00      | 3.10 ± .04      | 2.31 ± .00      | 2.50 ± .00      | 1.68 ± .00      |
> | Svd           | 2.46 ± .16       | 0.92 ± .01      | 1.94 ± .02      | 1.53 ± .03      | 2.24 ± .06      | 0.50 ± .00      | 3.67 ± .06      | 2.33 ± .01      |
> | Spectral      | 2.64 ± .11       | 0.91 ± .01      | 1.98 ± .04      | 1.46 ± .03      | 2.26 ± .09      | 0.41 ± .00      | 2.80 ± .01      | 2.13 ± .01      |
> | Mice          | 1.68 ± .05       | 0.77 ± .00      | 1.34 ± .01      | 1.16 ± .03      | 1.53 ± .04      | 0.20 ± .01      | 2.50 ± .00      | 1.16 ± .01      |
> | Knn           | 1.67 ± .02       | 0.72 ± .00      | 1.16 ± .03      | 0.95 ± .01      | 1.81 ± .03      | 0.10 ± .00      | 2.77 ± .01      | 1.38 ± .01      |
> | Gain          | 2.26 ± .11       | 0.86 ± .00      | 1.67 ± .03      | 1.23 ± .02      | 1.99 ± .03      | 0.46 ± .02      | 2.70 ± .00      | 1.31 ± .05      |
> | Miwae         | 2.37 ± .01       | 1.00 ± .00      | 1.81 ± .01      | 1.74 ± .04      | 2.79 ± .04      | 2.37 ± .00      | 2.57 ± .00      | 1.72 ± .00      |
> | Grape         | 1.46 ± .01       | 0.60 ± .00      | 0.75 ± .01      | 0.64 ± .01      | 1.36 ± .01      | 0.07 ± .00      | 2.50 ± .00      | 1.00 ± .00      |
> | Miracle       | 3.84 ± .00       | 0.70 ± .00      | 1.71 ± .05      | 3.12 ± .00      | 3.94 ± .01      | 0.18 ± .00      | 2.49 ± .00      | 1.13 ± .01      |
> | HyperImpute   | 1.76 ± .03       | 0.67 ± .01      | 0.84 ± .02      | 0.82 ± .01      | 1.32 ± .02      | 0.04 ± .00      | 2.58 ± .05      | 1.06 ± .01      |
> | M3-Impute     | 1.33 ± .04       | 0.60 ± .00      | 0.71 ± .01      | 0.59 ± .00      | 1.31 ± .01      | 0.06 ± .00      | 2.50 ± .00      | 0.99 ± .00      |
>
> (2) For the comment of using MSE instead of MAE in training, we would like to point out that we indeed used MSE for training, which was clearly mentioned in Section 3.5 of our manuscript.
>
> We acknowledge the efforts from all the reviewers and AC; however, we regret to know that these minor issues, which should be resolvable, were not raised during the discussion period.
>
> Sincerely,
>
> -- Authors